# Direct investigation of the reorientational dynamics of A-site cations in 2D organic-inorganic hybrid perovskite by solid-state NMR

Cheng-Chieh Lin [1,2], Shing-Jong Huang[3], Pei-Hao Wu[4], Tzu-Pei Chen[5], Chih-Ying Huang[1,2], Ying-Chiao Wang[5], Po-Tuan Chen[6], Denitsa Radeva[7], Ognyan Petrov[7], Vladimir M. Gelev[7], Raman Sankar [8], Chia-Chun Chen[9], Chun-Wei Chen [1,5,10 ✉] & Tsyr-Yan Yu [1,2,4 ✉]

Limited methods are available for investigating the reorientational dynamics of A-site cations in two-dimensional organic–inorganic hybrid perovskites (2D OIHPs), which play a pivotal role in determining their physical properties. Here, we describe an approach to study the dynamics of A-site cations using solid-state NMR and stable isotope labelling. $^2$H NMR of 2D OIHPs incorporating methyl-$d_3$-ammonium cations ($d_3$-MA) reveals the existence of multiple modes of reorientational motions of MA. Rotational-echo double resonance (REDOR) NMR of 2D OIHPs incorporating $^{15}$N- and $^{13}$C-labeled methylammonium cations ($^{13}$C,$^{15}$N-MA) reflects the averaged dipolar coupling between the C and N nuclei undergoing different modes of motions. Our study reveals the interplay between the A-site cation dynamics and the structural rigidity of the organic spacers, so providing a molecular-level insight into the design of 2D OIHPs.

[1] International Graduate Program of Molecular Science and Technology (NTU-MST), National Taiwan University, 10617 Taipei, Taiwan. [2] Molecular Science and Technology Program, Taiwan International Graduate Program (TIGP), Academia Sinica, 11529 Taipei, Taiwan. [3] Instrumentation Center, National Taiwan University, 10617 Taipei, Taiwan. [4] Institute of Atomic and Molecular Sciences, Academia Sinica, 10617 Taipei, Taiwan. [5] Department of Materials Science and Engineering, National Taiwan University, 10617 Taipei, Taiwan. [6] Department of Vehicle Engineering, National Taipei University of Technology, 10608 Taipei, Taiwan. [7] Department of Chemistry and Pharmacy, Sofia University, 1 James Bourchier Boulevard, 1164 Sofia, Bulgaria. [8] Institute of Physics, Academia Sinica, 115201 Taipei, Taiwan. [9] Department of Chemistry, National Taiwan Normal University, 11677 Taipei, Taiwan. [10] Center of Atomic Initiative for New Materials (AI-MAT), National Taiwan University, 10617 Taipei, Taiwan. ✉email: chunwei@ntu.edu.tw; tyyu@pub.iams.sinica.edu.tw

Organic–inorganic hybrid perovskites (OIHPs) have attracted a significant amount of attention for photovoltaic applications since their power conversion efficiency (PCE) has reached over 25%[1], which is already approaching the performance of commercial Si-based solar cells. The impressive PCE of OIHP photovoltaics is mainly attributed to their outstanding physical properties, such as a high optical absorption coefficient[2], low exciton binding energy[3] and long and balanced electron–hole diffusion length[4,5]. The chemical formula of OIHP is represented as $ABX_3$ where A is an organic cation, such as methylammonium ($MA^+$), B is a divalent metal cation, such as lead(II) ion ($Pb^{2+}$) and X is a halide anion[6,7]. Although the electronic structure of OIHP near the band edges is mainly determined by the inorganic Pb and halide ions, A-site organic cations have also been reported to play an important role in affecting the device performance and stability, instead of only acting as a passive component for a charge compensation for the $[PbI_3]^-$ lattice[8–13]. For example, it has been proposed that the molecular rotations of A-site cations may cause distortion of the $PbI_6$ octahedral unit, the consequent dynamical change of the band structure being at the origin of the slow carrier recombination and the superior conversion efficiency of $CH_3NH_3PbI_3$[14]. Several experimental techniques have revealed the dynamics of A-site cations, including solid-state nuclear magnetic resonance (ssNMR)[10,15,16], neutron powder diffraction (NPD)[17] and inelastic neutron scattering[18,19]. In particular, ssNMR has emerged as a useful tool for studying OIHP[20] and the cation reorientational dynamics in OIHP. The interplay between the charge carrier lifetimes and the reorientational dynamics of A-site cations in OIHP photovoltaics has been providing evidence of the polaronic nature of charge carriers in perovskite photovoltaics[10,15,21].

Recently, two-dimensional organic–inorganic hybrid perovskites (2D OIHPs) have attracted great attention owing to their superior ambient stability, and promising optoelectronic properties[22–25]. Ruddlesden–Popper perovskites are a typical example of layered 2D organic–inorganic hybrid perovskites having the generic chemical formula $A'_2A_{n-1}M_nX_{3n+1}$. In this formula, A′ represents an organic spacer, such as long-chain alkylammonium cation (e.g. 1-butylammonium, $BA^+$) or phenyl alkylammonium cation (e.g. 2-phenethylammonium, $PEA^+$), A is an organic cation, M is a metal, X is a halide and $n$ is the number of octahedral slabs per unit cell[26–29]. Layered 2D OIHP consists of a self-assembled periodic array of inorganic perovskite layers of corner-sharing $PbX_6$ octahedral slabs, separated by the organic spacers in the lattice framework[30,31]. Accordingly, they exhibit a naturally formed "multiple quantum-well" (MQW) structure. The semiconducting inorganic $PbX_6$ perovskites, with a smaller bandgap, act as potential "wells", while the insulating organic layers, possessing a larger bandgap, act as potential "barriers". The value of $n$ is the number of inorganic octahedral slabs per unit cell that determines the width of the QW[32–34]. Layered 2D OIHPs have emerged as a new class of outstanding optoelectronic materials due to their unique tunable physical properties, and structural flexibility, achieved by controlling the value of $n$ and the thickness of the perovskite slabs[35–37]. Because layered 2D OIHPs incorporate organic spacers between the flexible inorganic layers, they exhibit a greater structural versatility compared to their 3D OIHP counterparts[38]. It has been reported that the manipulations of the A-site cations and organic spacers may cause the structural rearrangement, or deformation, of the $PbI_6$ octahedral unit in 2D OIHPs, which may further influence their electronic band structures near the band edges, and the corresponding optical and electronic behaviours[39–41]. Unlike their 3D OIHP counterparts consisting of only A-site cations, the detection of reorientation dynamics of A-site cations in 2D OIHPs with $n \geq 2$ by ssNMR is a challenging task due to signal overlap with the additional organic spacers. Subsequently, the conventional $^1H$

and $^{14}N$ NMR relaxation methods for characterising the dynamics of cations in 3D OIHP[10,15,16] are not applicable for studying the dynamics of A-site cations in 2D OIHPs. Until now, only the dynamics of the organic spacers at the 2D OIHP crystals have been analysed based on these ssNMR methods[29,41–44]. Here, we employed isotope labelling to distinguish A-site from spacer cations and avoid spectral overlap in 2D OIHPs with $n = 2$. Specifically, we employed $^{13}C,^{15}N$-MA and $CD_3NH_3^+$ as A-site cations, and investigated the dynamics of A-site molecular cations by REDOR NMR and $^2H$ NMR, respectively.

To study the dynamics of MA, we first applied two-dimensional $^{13}C$–$^{15}N$ correlation double cross-polarisation magic-angle spinning spectroscopy[45], 2D ($^{13}C,^{15}N$) DCP MAS, to observe the NMR signal of MA with a substantial sensitivity enhancement and without the interference of the signal of the spacer cations. REDOR NMR and $^2H$ NMR were further employed to study the reorientational dynamics of MA in 2D OIHPs, which provides motional average to the dipolar coupling between the $^{13}C$ and $^{15}N$ nuclei of $^{13}C,^{15}N$-MA and to the deuterium quadrupole coupling of $CD_3NH_3^+$, respectively. Accordingly, both the environments and the dynamics of A-site molecular cations can be revealed. We further characterised the structural and optoelectronic properties of 2D OIHPs using powder X-ray diffraction spectroscopy (PXRD), absorption spectroscopy and photoluminescence spectroscopy (PL), in addition to $^{13}C$ CPMAS NMR characterisation of the incorporated spacer molecules at the various temperatures. Our PXRD and PL results indicated that the choice of the spacer could influence the structures and the optoelectronic properties of 2D OIHPs, as mentioned in the literature[33,39,41]. We further showed that a 1-butylammonium spacer ($BA^+$) is less rigid than a 2-phenethylammonium spacer ($PEA^+$) according to our $^{13}C$ CPMAS NMR results. Finally, the detection of the reorientational dynamics of MA by the two ssNMR methods reported here allows us to further examine the interplay between the rigidity of organic spacers and the dynamics of the A-site cations in 2D OIHPs. The present study complements previous NMR studies focusing on the spacer molecules or the frameworks[29,41–44], and should provide insights into the future design of 2D OIHP materials.

## Results

**Dynamics of A-site cations of 2D OIHPs.** We compared two types of 2D OIHP, one containing the 1-butylammonium spacer (2D $(BA)_2MAPb_2I_7$ ($n = 2$)) and one containing the 2-phenethylamine spacer (2D $(PEA)_2MAPb_2I_7$ ($n = 2$)). Both were synthesised using the slow evaporation at a constant-temperature (SECT) growth method[28]. For comparison, we also synthesised 3D $MAPbI_3$, in which no organic spacer is present. The perovskites were synthesised using methylammonium iodide (MAI) that was $^{13}C,^{15}N$-labeled, methyl-$d_3$-labeled or natural abundance.

$^{13}CH_3^{15}NH_3I$ ($^{13}C,^{15}N$-MAI) was synthesised from commercially available $^{13}CH_3I$ and $^{15}N$-phthalimide following the classic Gabriel synthesis (Supplementary Methods). The 2D ($^{13}C,^{15}N$) DCP MAS NMR of the 2D OIHP crystals synthesised with $^{13}C,^{15}N$-methylammonium iodide exhibited signal solely from the dipolar-coupled $^{13}C$–$^{15}N$ spin pair, which is practically absent (comprising 0.004%) from a natural-abundance sample. The purity of the 2D OIHP samples was examined by PXRD spectroscopy, which confirmed the single-crystal structure of 2D OIHP (Supplementary Fig. 1). The PL spectra of the natural-abundance and $^{13}C,^{15}N$ materials were nearly identical as shown in Supplementary Fig. 2. Thus, the structural and the physical properties of 2D OIHPs were not

noticeably altered by the $^{13}C$ and $^{15}N$ nuclei. As shown in Fig. 1b, the natural-abundance methylamine (MA) signal is poor and largely overlapped with the C2 signal of BA. In contrast, the 2D ($^{13}C$,$^{15}N$) DCP MAS spectra of the $^{13}C$,$^{15}N$-labeled 2D OIHP allow observation of the $^{13}C$–$^{15}N$ spin pair free of interference from the signals of the spacer cations (Fig. 1c). This also enabled the unambiguous assignment of the resonance peaks of the unlabeled MA in Fig. 1b. Notably, the 2D ($^{13}C$,$^{15}N$) DCP MAS spectrum of 2D $(BA)_2(^{13}C,^{15}N\text{-MA})Pb_2I_7$ ($n = 2$) contains two cross-peaks (Fig. 1c), suggesting the existence of the two different local environments of the A-site MA. The major component corresponds to a 31.4 ppm peak, while the minor component is associated with a 27.2 ppm peak in the $^{13}C$ direct excitation MAS spectrum (Fig. 1b). Comparison of the $^{13}C$ NMR spectra of the $^{13}C$,$^{15}N$-perovskite and the $^{13}C$,$^{15}N$-methylammonium iodide precursor (Supplementary Fig. 3) ruled out the possibility of the minor MA component being the unreacted $^{13}C$,$^{15}N$-methylammonium iodide precursor. Moreover, the PXRD and UV data (Supplementary Figs. 1 and 2) showed one series of periodic repetitions of Miller planes and one excitonic absorption peak in both the $^{13}C$,$^{15}N$-labeled and the unlabeled 2D $(BA)_2MAPb_2I_7$ ($n = 2$). Thus, the minor MA spectral component indicates the presence of minor structural impurity, as suggested in a recent work[46]. The amount of the minor MA component was estimated to be roughly 3% of the total amount of MA based on the ratio of the peak intensities of the $^{13}C$ direct excitation MAS spectrum. In contrast, only one local environment of MA in 2D $(PEA)_2(^{13}C,^{15}N\text{-MA})Pb_2I_7$ ($n = 2$) was observed in Fig. 1c. Interestingly, the intensity of the cross peak of the major MA component of 2D $(BA)_2(^{13}C,^{15}N\text{-MA})Pb_2I_7$ ($n = 2$) is less than that of the minor MA component, suggesting that the major MA component is associated with the lower transfer efficiency of double cross-polarisation (DCP). The reason for the lower DCP transfer efficiency is attributed to the presence of the reorientational motion of the C–N bond of MA. In summary, different local environments and the dynamics of the A-site cations were observed in 2D OIHP crystals incorporated with different organic spacers.

The reorientation dynamics of the MA cation can be studied in a quantitative manner by $^{13}C\{^{15}N\}$REDOR NMR, which measures the time evolution of the internuclear dipolar interaction. The pulse sequence is shown in Fig. 2a, where $^{13}C$ is the observed spin and $^{15}N$ is the dephasing spin. Two sets of the interleaved experimental REDOR signals were recorded. The full echo signal, denoted as $S_0$, was recorded without the dephasing pulses and the reduced echo signal, denoted as S, was recorded with the dephasing pulses switched on to recouple the dipolar interaction of the $^{13}C$–$^{15}N$ spin pairs. The data were plotted as a REDOR dephasing curve of $\Delta S/S_0$ versus different dephasing times $N \times Tr$, where $Tr$ is the rotor period and N is $2(n + 1)$. In a typical solid-state powder, in the absence of reorientational motion, the distance-dependent nature of the REDOR dephasing curve allows for the extraction of internuclear distances at angstrom resolution[47]. A powder sample of $^{13}C$,$^{15}N$-MAI crystals was used as a reference, where the reorientational motion of the $^{13}C$–$^{15}N$ vector is absent. Fitting the experimental $^{13}C\{^{15}N\}$REDOR results (orange open circles) to a simulated dephasing curve (orange dashed line) indicate a C–N bond length of 1.51 Å, consistent with published single-crystal X-ray crystallography bond lengths of 1.469–1.516 Å[33,48–51].

While REDOR NMR is a popular method for measuring the internuclear distance at atomic resolution, it can also be applied to study the molecular motion of a dipolar-coupled nuclear spin pair[52,53]. Using a model-free approach[54,55], the theoretical analysis of REDOR dephasing for a $^{13}C$–$^{15}N$ spin pair undergoing reorientational motion of the $^{13}C$–$^{15}N$ vector is described in Supplementary Methods. An order parameter $\mathscr{S}$ is used to describe the motional modulation of the effective dipolar interaction[54,55]. For the case of a rigid $^{13}C$–$^{15}N$ vector, in the absence of reorientational motion, $\mathscr{S} = 1$ and the REDOR dephasing maximum appears around 2 ms of the dephasing time. In contrast, no REDOR dephasing will be observed if the $^{13}C$–$^{15}N$ vector is undergoing completely random reorientational motion ($\mathscr{S} = 0$). In the case of $0 < \mathscr{S} < 1$, a reduced REDOR dephasing value around 2 ms of the dephasing time will be obtained, as indicated in Supplementary Fig. 4. Experimentally, we chose to compare the $^{13}C\{^{15}N\}$REDOR dephasing values ($\Delta S/S_0$) at 2.4 ms of the dephasing time, denoted as $(\Delta S/S_0)_{2.4ms}$, of the various samples so that the xy-8 phasing cycling could be applied to the dephasing pulses. Given that 10 kHz of the MAS frequency was used in this series of REDOR experiments, a reduced $^{13}C\{^{15}N\}$ REDOR $(\Delta S/S_0)_{2.4ms}$ value indicated that the presence of the reorientational motion of the C–N vector partially averaged out the $^{13}C$–$^{15}N$ dipole-dipole interaction over the period of 100 μs. In summary, the degree of the motional averaging caused by the reorientational motion of the C–N vectors can be characterised by an order parameter $\mathscr{S}$ and can be directly associated with the reduction of the $^{13}C\{^{15}N\}$ REDOR $(\Delta S/S_0)_{2.4ms}$ values.

We systematically studied the reorientational dynamics of $^{13}C$,$^{15}N$-MA incorporated in three different perovskite crystals by measuring the effective $^{13}C$–$^{15}N$ dipolar interactions of $^{13}C$,$^{15}N$-MA using $^{13}C\{^{15}N\}$REDOR NMR. While the $(\Delta S/S_0)_{2.4ms}$ value of the powder sample of $^{13}C$,$^{15}N$-MAI crystals (rigid $^{13}C$–$^{15}N$ vector, $\mathscr{S} = 1$), was around the maximum value of 1, reduced $(\Delta S/S_0)_{2.4ms}$ values were observed for the three perovskite samples: 2D $(PEA)_2(^{13}C,^{15}N\text{-MA})Pb_2I_7$ ($n = 2$), 2D $(BA)_2(^{13}C,^{15}N\text{-MA})Pb_2I_7$ ($n = 2$), and 3D $(^{13}C,^{15}N\text{-MA})PbI_3$ (Fig. 2(c)–(f)). No organic spacer is present in the 3D $(^{13}C,^{15}N\text{-MA})PbI_3$ which was included for comparison. The minor MA peak of 2D $(BA)_2(^{13}C,^{15}N\text{-MA})Pb_2I_7$ ($n = 2$) was fully dephased at 2.4 ms indicating the absence of the reorientational motion of MA. In contrast, the $(\Delta S/S_0)_{2.4ms}$ values observed for 3D $(^{13}C,^{15}N\text{-MA})PbI_3$ and the major MA peak in 2D $(BA)_2(^{13}C,^{15}N\text{-MA})Pb_2I_7$ ($n = 2$) were less than 10%, indicating that MA cations in these two samples underwent reorientational motion. Interestingly, the $(\Delta S/S_0)_{2.4ms}$ value measured for the 2D $(PEA)_2(^{13}C,^{15}N\text{-MA})Pb_2I_7$ sample was greater than the $(\Delta S/S_0)_{2.4ms}$ values for 2D $(BA)_2(^{13}C,^{15}N\text{-MA})Pb_2I_7$ by more than 10%. Thus, the A-site cations undergo more restricted reorientational motion in 2D $(PEA)_2(^{13}C,^{15}N\text{-MA})Pb_2I_7$ than in $(BA)_2(^{13}C,^{15}N\text{-MA})Pb_2I_7$.

To explore the spacer-dependent dynamics of the A-site cation, the REDOR $(\Delta S/S_0)_{2.4ms}$ values of $^{13}C\{^{15}N\}$REDOR experiments were recorded at various temperatures. As plotted in Fig. 3, only minor change of the REDOR $(\Delta S/S_0)_{2.4ms}$ values were measured for 3D $(^{13}C,^{15}N\text{-MA})PbI_3$ during the cooling process from 308 to 243 K, indicating that the reorientational motion of the A-site MA did not change significantly in this temperature range. By contrast, there was an obvious change of the REDOR $(\Delta S/S_0)_{2.4ms}$ values of 2D $(BA)_2(^{13}C,^{15}N\text{-MA})Pb_2I_7$ ($n = 2$) which were found to increase from 0.06 at 308 K, to 0.43 at 243 K, implying that the reorientational motion of MA becomes more restricted as the temperature decreases. In contrast, the reorientational motion of MA in 2D $(PEA)_2(^{13}C,^{15}N\text{-MA})Pb_2I_7$ ($n = 2$) changed only slightly during cooling, as seen from the slightly increased REDOR $(\Delta S/S_0)_{2.4ms}$ value.

Lineshape analysis of $^2H$ NMR spectra can yield useful information about the motion of the deuterium quadrupole in solid. $^2H$ NMR spectra of 2D $(PEA)_2(d_3\text{-MA})Pb_2I_7$ ($n = 2$) and

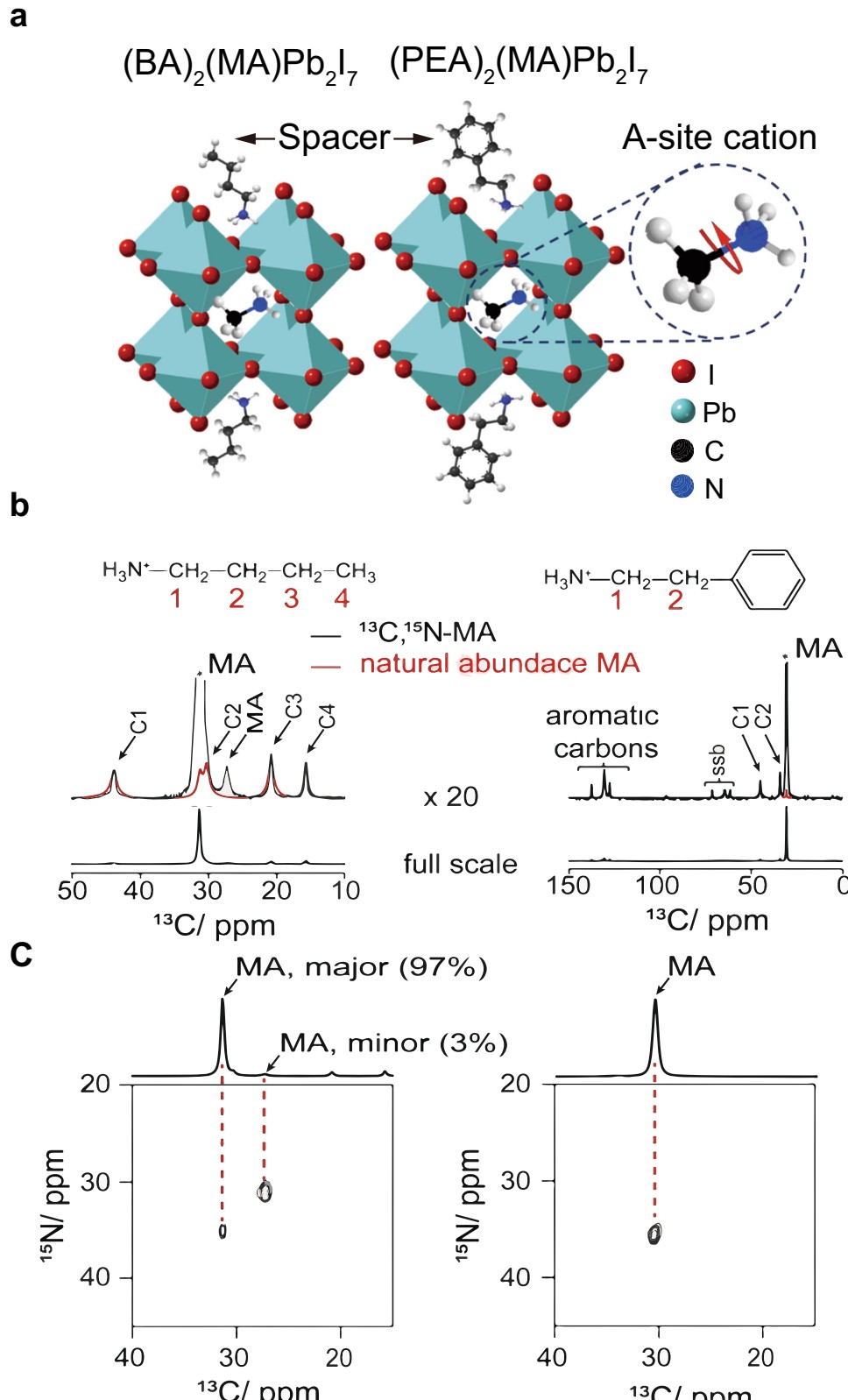

**Fig. 1 $^{13}$C NMR spectral characterisation of 2D OIHP crystals ($n = 2$). a** The structural models of 2D (BA)$_2$(MA)Pb$_2$I$_7$ ($n = 2$) and 2D (PEA)$_2$(MA)Pb$_2$I$_7$ ($n = 2$). **b** The $^{13}$C CPMAS spectra of 2D (BA)$_2$(MA)Pb$_2$I$_7$ ($n = 2$) and 2D (PEA)$_2$(MA)Pb$_2$I$_7$ ($n = 2$) synthesised with $^{13}$C,$^{15}$N-MA, respectively. The spectra with ×20 magnification are shown on the top, overlayed with the spectra of the materials synthesised with natural-abundance MA. The two sets of spectra have been normalised by the height of the C1 carbon peak of BA and the aromatic carbon peak at 130.7 ppm, respectively. **c** 2D DCP MAS ($^{13}$C,$^{15}$N) correlation spectra of 2D (BA)$_2$($^{13}$C,$^{15}$N-MA)Pb$_2$I$_7$ ($n = 2$) and 2D (PEA)$_2$($^{13}$C,$^{15}$N-MA)Pb$_2$I$_7$ ($n = 2$), respectively. The $^{13}$C direct excitation MAS NMR spectra are overlayed on the top.

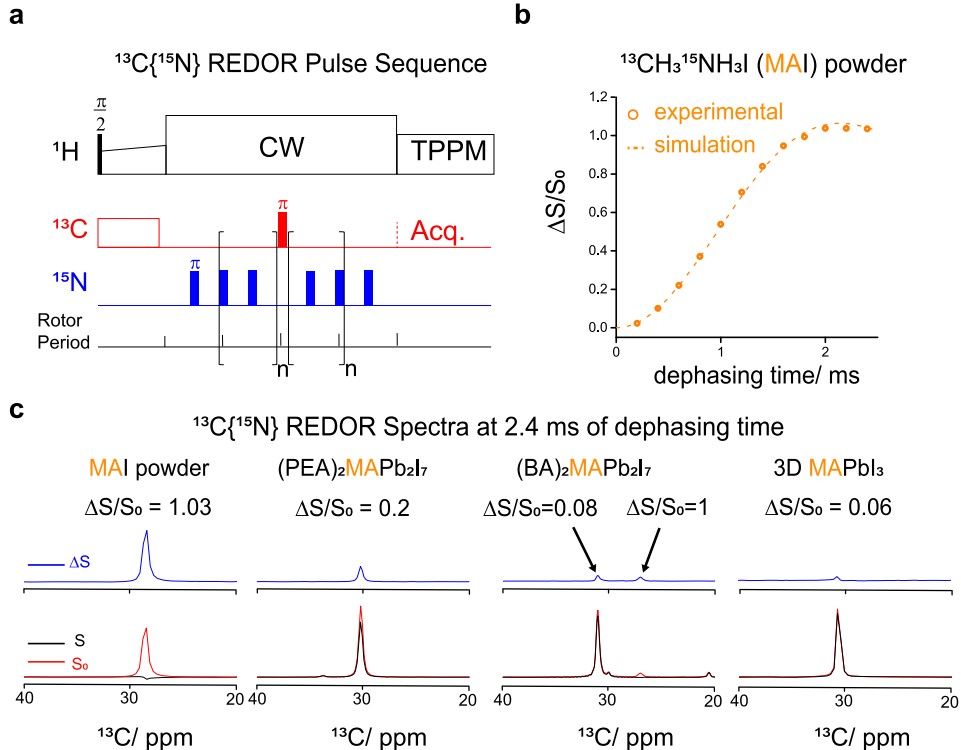

**Fig. 2 $^{13}$C{$^{15}$N}REDOR NMR characterisation. a** $^{13}$C{$^{15}$N}REDOR NMR pulse sequence. **b** The experimental {$^{13}$C}$^{15}$N REDOR dephasing curve of the precursor $^{13}$C,$^{15}$N-MAI powder. The dashed line is the REDOR curves simulated using the length of carbon-nitrogen bond of MA molecule reported in X-ray crystallography research[33, 48–51]. **c** The {$^{13}$C}$^{15}$N REDOR spectra at the 2.4 ms dephasing time of $^{13}$C,$^{15}$N-MAI powder, 2D (PEA)$_2$($^{13}$C,$^{15}$N-MA)Pb$_2$I$_7$ ($n = 2$), 2D (BA)$_2$($^{13}$C,$^{15}$N-MA)Pb$_2$I$_7$ ($n = 2$) and 3D ($^{13}$C,$^{15}$N-MA)PbI$_3$.

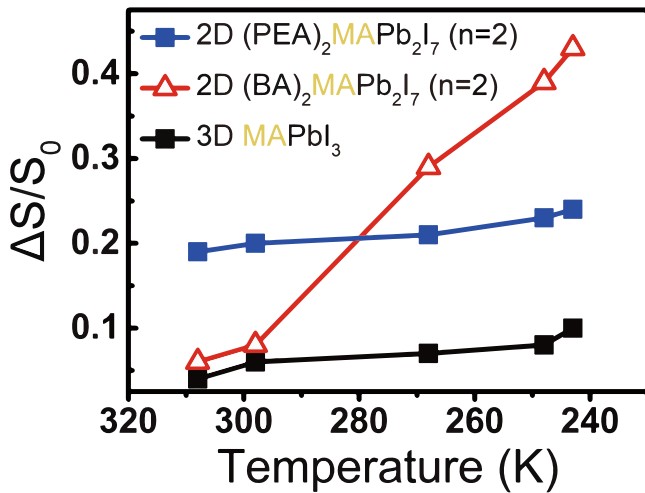

**Fig. 3 The temperature-dependent $^{13}$C{$^{15}$N}REDOR dephasing (ΔS/S$_0$) of $^{13}$C,$^{15}$N-MA incorporated in different materials.** The ΔS/S$_0$ of MA peak in 2D (PEA)$_2$($^{13}$C,$^{15}$N-MA)Pb$_2$I$_7$ ($n = 2$) (blue solid squares), 2D (BA)$_2$($^{13}$C,$^{15}$N-MA)Pb$_2$I$_7$ ($n = 2$) (red open triangles)* and 3D ($^{13}$C,$^{15}$N-MA)PbI$_3$ (black solid squares) at the dephasing time of 2.4 ms. *The data were obtained from the major MA peak of 2D (BA)$_2$($^{13}$C,$^{15}$N-MA)Pb$_2$I$_7$ ($n = 2$).

2D (BA)$_2$(d$_3$-MA)Pb$_2$I$_7$ ($n = 2$) were acquired at 11.7 T in the 243–298 K range. For a non-rotating CD$_3$ group, the quadrupole splitting ($\nu_Q$), measured cusp to cusp of the $^2$H spectrum, would be ~120 kHz. Rapid C3 rotation would reduce this $\nu_Q$ value threefold to ~40 kHz. The existence of additional reorientational motion of the C3 axis would result in a narrower spectral lineshape[11]. Interestingly, multiple $\nu_Q$ values were observed from the $^2$H NMR spectra of both 2D OIHPs (Fig. 4), suggesting the existence of multiple modes of reorientational motions. It worth mentioning that there was only one chemical environment observed for the MA in 2D (PEA)$_2$($^{13}$C,$^{15}$N-MA)Pb$_2$I$_7$ ($n = 2$) as indicated in Fig. 1c. The existence of multiple modes of the C–N reorientational motion was also suggested previously in 3D MAPbI$_3$ perovskite crystals using neutron powder diffraction[17], quasielastic neutron scattering measurement[18] and first-principle calculations[56]. The orientation of the C–N vector and the geometry of its motion with respect to the molecular frame of 2D perovskite crystals are needed to construct a detailed model for quantitative analysis of the $^2$H NMR spectra. However, these parameters are not currently available in the literature. Nonetheless, we can still learn about the reorientational motion of the C–N vector based on qualitative analysis. The $^2$H NMR spectral lineshape of 2D (BA)$_2$(d$_3$-MA)Pb$_2$I$_7$ ($n = 2$) was shown to widen significantly as the temperature was decreased from 298 to 243 K, indicating that the reorientational motion of the C–N vector changed significantly as the temperature was decreased. This result is consistent with our REDOR result, showing the significant change of the $^{13}$C{$^{15}$N} REDOR (ΔS/S$_0$)$_{2.4ms}$ values of 2D (BA)$_2$($^{13}$C,$^{15}$N-MA)Pb$_2$I$_7$ during cooling. In contrast, no obvious change of the $^2$H NMR spectral lineshape of 2D (PEA)$_2$(d$_3$-MA)Pb$_2$I$_7$ was observed during cooling. This finding echoes the REDOR data for 2D (PEA)$_2$($^{13}$C,$^{15}$N-MA)Pb$_2$I$_7$. Overall, we observed that the choice of the spacer cation in 2D perovskite affects the temperature dependence of the reorientational motional of MA. It is worth mentioning that full REDOR dephasing at 2.4 ms was observed for 3% (the minor component) of MA in 2D (BA)$_2$(d$_3$-MA)Pb$_2$I$_7$, indicating the absence of the reorientational motion of the C–N vector. The quadrupole splitting ($\nu_Q$) associated with the minor MA component in 2D

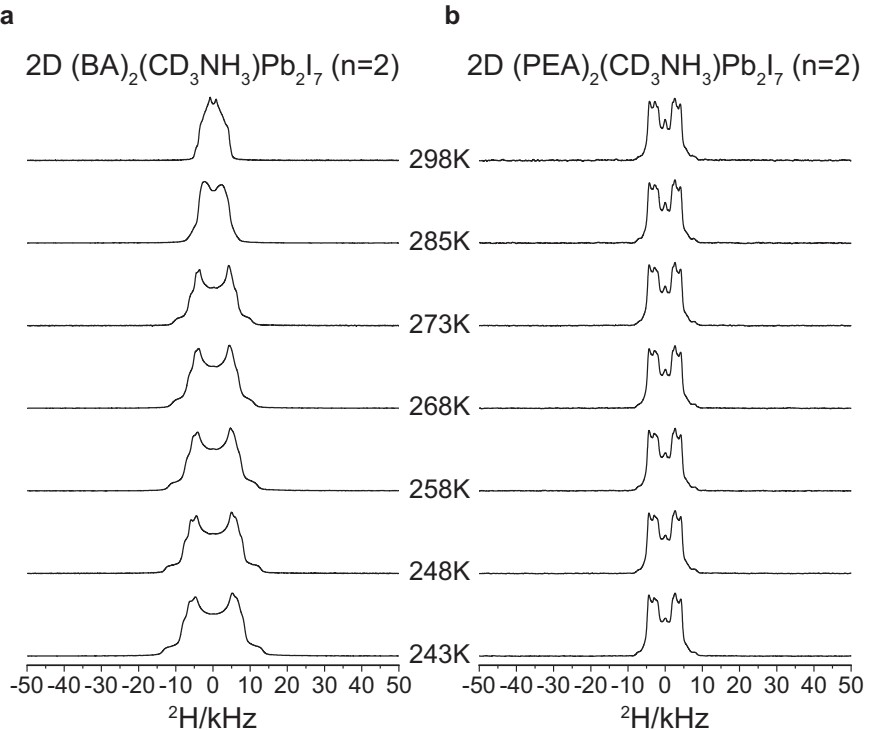

**a** 2D (BA)$_2$(CD$_3$NH$_3$)Pb$_2$I$_7$ (n=2)   **b** 2D (PEA)$_2$(CD$_3$NH$_3$)Pb$_2$I$_7$ (n=2)

**Fig. 4 $^2$H NMR spectra of 2D OIHP crystals (*n* = 2).** The full-scale $^2$H spectra of **a** 2D (BA)$_2$(d$_3$-MA)Pb$_2$I$_7$ (*n* = 2) and **b** 2D (PEA)$_2$(d$_3$-MA)Pb$_2$I$_7$ (*n* = 2) recorded at various temperatures, ranging from 298 to 243 K.

(BA)$_2$(d$_3$-MA)Pb$_2$I$_7$ should therefore be equal or greater than 40 kHz, which was not observed (Fig. 3a). This discrepancy may be due to the low sensitivity of the deuterium spectra, compared to that of the $^{13}$C spectra of $^{13}$C,$^{15}$N-MA.

**$^{13}$C CPMAS NMR, PXRD and DSC Characterisations.** To further investigate the response of the spacer cations during cooling, the $^{13}$C CPMAS NMR spectra of 2D OIHPs were recorded at various temperatures. Figure 5a, b presents the magnified $^{13}$C CPMAS NMR spectra, highlighting the resonance peaks of the spacer cations of the two 2D OIHPs. An obvious conformational change of the 1-butylammonium (BA$^+$) cation spacer in 2D (BA)$_2$($^{13}$C,$^{15}$N-MA)Pb$_2$I$_7$ (*n* = 2) was observed, where the $^{13}$C resonances of C3 and C4 of BA are split into two peaks, during the cooling process from 268 to 243 K. The peak splitting shows the existence of two BA conformations in this temperature range, indicating the packing geometry of the BA molecules was altered during cooling. In contrast, neither peak splitting nor peak position changes were found in the $^{13}$C CPMAS NMR spectrum of 2D (PEA)$_2$($^{13}$C,$^{15}$N-MA)Pb$_2$I$_7$ (*n* = 2) during the cooling process, indicating no change in the conformation and the chemical environment of PEA. A signal reduction of a phenyl carbon resonance peak at 130 ppm with increasing temperature may be due to the flipping of the phenyl ring. The full-scale $^{13}$C CPMAS NMR spectra which highlight the resonance peaks of the molecular cation MA, are also shown in Supplementary Fig. 5. There is no obvious change of the peak position nor intensity for the $^{13}$C resonance peak of MA for the PEA-based 2D OIHPs. In contrast, the major peak of MA in 2D (BA)$_2$($^{13}$C,$^{15}$N-MA)Pb$_2$I$_7$ (*n* = 2) displays a clear up-field shift and significant intensity increase on cooling from 298 to 268 K, suggesting that a change in the MA chemical environment accompanied the change of the packing geometry of the BA ions. Differential scanning calorimetry (DSC) and PXRD were further used to study the structural response of 2D (BA)$_2$(MA)Pb$_2$I$_7$

(*n* = 2) and 2D (PEA)$_2$(MA)Pb$_2$I$_7$ (*n* = 2) during the cooling process. The endothermic peak observed in the DSC measurement of 2D (BA)$_2$(MA)Pb$_2$I$_7$ (*n* = 2) indicated a clear phase change occurring at ~280 K (Supplementary Fig. 6). This echoes the finding in a recent study[33], showing the associated phase change was from Cmcm to P-1 space group. In contrast, no evidence of phase change was found in the DSC measurement of (PEA)$_2$(MA)Pb$_2$I$_7$ (*n* = 2). Moreover, based on the comparison of PXRD patterns recorded at 300 K and 250 K (Fig. 5c, d) a clear structural change for (BA)$_2$(MA)Pb$_2$I$_7$ (n = 2) was observed with cooling from 300 to 250 K, while no change was observed for the (PEA)$_2$(MA)Pb$_2$I$_7$ (*n* = 2).

**PL characterisation.** It is well-known that the structural deformation of the octahedral layers, induced by changing the packing geometry of the organic spacers, may strongly affect the optical and electronic properties of 2D OIHPs[33,39,41]. Photoluminescence spectroscopy (PL) during the cooling from 300 to 250 K revealed a significant blue shift from 581 to 574 nm in 2D (BA)$_2$($^{13}$C,$^{15}$N-MA)Pb$_2$I$_7$ (*n* = 2), as shown in Supplementary Fig. 7. The corresponding PL emission peak of 2D (PEA)$_2$($^{13}$C,$^{15}$N-MA)Pb$_2$I$_7$ (*n* = 2) remained unshifted. The results suggest that the phase change or structural deformation of 2D OIHPs induced by changing the packing geometry of the organic spacers, may strongly affect the optical properties of 2D OIHPs.

**The role of the rigidity of spacer cation in 2D OIHPs.** These results may help to elucidate the important role of the organic spacers in determining the structure and properties of 2D OIHPs. Flexible alkyl-chain cations such as 1-butylammonium (BA) produce a more fluctuating structure at ambient conditions compared to 2D OIHPs based on 2-phenethylammonium (PEA) cations, consistent with recent theoretical results[38]. The CH-π stacking between the aromatic rings of the PEA spacer may restrict thermal motions between two inorganic perovskite layers. By contrast, there

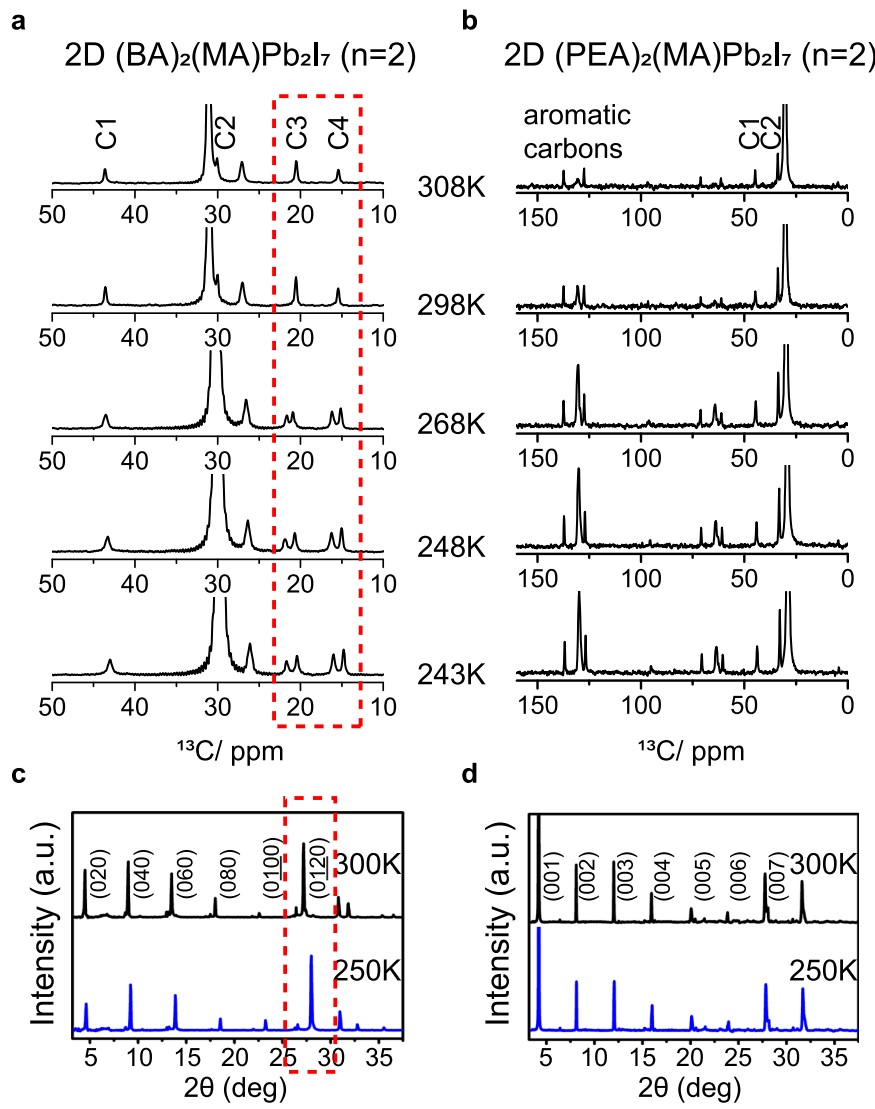

**Fig. 5 The $^{13}C$ CPMAS spectra and PXRD spectra of 2D OIHP crystals ($n = 2$).** The $^{13}C$ CPMAS spectra of **a** 2D $(BA)_2(^{13}C,^{15}N\text{-}MA)Pb_2I_7$ ($n = 2$) and **b** 2D $(PEA)_2(^{13}C,^{15}N\text{-}MA)Pb_2I_7$ ($n = 2$) recorded at various temperatures, ranging from 308 to 243 K. The PXRD patterns of **c** 2D $(BA)_2MAPb_2I_7$ ($n = 2$) and **d** 2D $(PEA)_2MAPb_2I_7$ ($n = 2$) measured at 300 K (black colour) and 250 K (blue colour). The red dashed boxes highlight the obvious changes of the $^{13}C$ CPMAS and PXRD spectra during cooling.

is no such interaction among the alkyl BA cations, which may result in more flexibility and more structural dynamics in the inorganic perovskite layers. Accordingly, the 2D OIHP crystal, consisting of PEA organic spacers, exhibits a relatively rigid structure with lower conformational freedom of the crystal structure, as compared to its BA counterpart. That temperature-induced structural change of 2D OIHPs with cooling is mainly attributed to the influence of the organic spacers can also be concluded from the $^{13}C$ CPMAS NMR spectra of 2D OIHP ($n = 1$), which lack A-site cations (Supplementary Fig. 8). For 2D OIHPs consisting of BA spacers, the structural deformation of the inorganic $PbI_6$ octahedral layers with cooling was induced by changing the packing geometry of BA, as seen from the ssNMR, PXRD and DSC data. In contrast, there was no such structural deformation of the inorganic octahedral layers in the 2D OIHPs consisting of PEA spacers with cooling in this temperature range.

## Discussion
Here, we used two different ssNMR methods to study the reorientational motions of the A-site cation MA incorporated in 2D

OIHPs with $n \geq 2$. The reorientational motion of the C–N bond of MA modulates the effective dipolar coupling between the $^{13}C$ and $^{15}N$ nuclei of $^{13}C,^{15}N\text{-}MA$, which can be studied using REDOR NMR. The same reorientational motion of the C–N bond of MA also modulates the quadrupolar splitting ($\nu_Q$) of the $^2H$ NMR spectra of $CD_3NH_3^+$ incorporated in 2D OIHPs. Thus, the $^2H$ NMR lineshape also reflects the reorientational dynamics of the C–N vector.

REDOR NMR analyses were performed on $^{13}C,^{15}N\text{-}MAI$ crystal powder, 2D $(BA)_2(^{13}C,^{15}N\text{-}MA)Pb_2I_7$ ($n = 2$), 2D $(PEA)_2(^{13}C,^{15}N\text{-}MA)Pb_2I_7$ ($n = 2$) and 3D $(^{13}C,^{15}N\text{-}MA)PbI_3$, to compare the reorientational dynamics of $^{13}C,^{15}N\text{-}MA$ incorporated in different materials. The reorientational motion of MA is absent for the $^{13}C,^{15}N\text{-}MAI$ crystal powder sample, evident from the consistency between the experimental REDOR curve and the simulated REDOR curve using the carbon-nitrogen bond length obtained in single-crystal X-ray studies. The $^{13}C,^{15}N\text{-}MA$ in 2D $(PEA)_2(^{13}C,^{15}N\text{-}MA)Pb_2I_7$ ($n = 2$) was found to undergo a more restricted reorientational motion at room temperature when compared to 2D $(BA)_2(^{13}C,^{15}N\text{-}MA)Pb_2I_7$ ($n = 2$) and 3D $(^{13}C,^{15}N\text{-}MA)PbI_3$. Moreover, an obvious change of the

reorientational dynamics of $^{13}C,^{15}N$-MA in 2D $(BA)_2(^{13}C,^{15}N$-MA)$Pb_2I_7$ ($n = 2$) occurred during cooling from 298 to 268 K, evident from an obvious increase in the $^{13}C\{^{15}N\}$ REDOR ($\Delta S/S_0)_{2.4ms}$ value. It is worth mentioning that the significant change in the reorientational dynamics can also be correlated to the significant change of the chemical environment of MA, evident from the clear up-field shift and a significant increase in the intensity of the $^{13}C$ CPMAS resonance peak of MA during cooling from 298 to 268 K. In contrast, only slight changes of MA reorientational dynamics were observed in 2D $(PEA)_2(^{13}C,^{15}N$-MA)$Pb_2I_7$ ($n = 2$) and 3D ($^{13}C,^{15}N$-MA)$PbI_3$ with cooling. Accordingly, the change of the dynamics of the A-site cations in response to temperature change depends on the choice of the spacer cations of 2D OIHPs. The results of the temperature variation $^2H$ NMR lineshape analyses of $d_3$-MA incorporated in 2D OIHPs are consistent with the REDOR NMR results, showing that the choice of the spacer cations in 2D OIHPs may affect the reorientational dynamics of the A-site cations. It is important to note that $^2H$ NMR lineshape analyses further revealed the existence of the multiple modes of the reorientational motions of MA. Thus, we should interpret the dipolar coupling measured in our REDOR measurements as the average dipolar coupling between the $^{13}C$ and $^{15}N$ nuclei of MA undergoing different modes of motions, which are characterised by different values of the order parameters. We recorded both $^{13}C$ CPMAS NMR and PXRD at different temperatures to investigate the structural changes of the organic spacer cations and of the inorganic frameworks of 2D OIHPs with cooling, respectively. 2D BA-based OIHP exhibits less structural rigidity than 2D PEA-based OIHP. The structural deformation of the octahedral perovskite layers induced by the conformational change of the alkyl BA spacers in the 2D BA-based OIHPs occurs with cooling[33]. In contrast, there is no such structural change in both the inorganic framework of perovskites and the conformation of organic spacers in the 2D PEA-based OIHPs, evident from the ssNMR and PXRD analyses, respectively. Consequently, the structural deformation of the inorganic octahedral layers induced by changing the packing geometry of the BA organic spacers occurs on cooling, resulting in the change of the reorientational motion of the A-site cations and the change of optoelectronic property indicated by the shift in the PL spectrum. When the rigid spacer cation PEA was incorporated in 2D OIHP crystals, there is no such significant change neither in the conformation of the spacer cation nor in the inorganic frameworks with cooling. Consequently, the reorientational dynamics of the A-site cation were found to stay relatively unchanged in the PEA-containing 2D OIHPs. In conclusion, the ssNMR study provides information on the reorientational dynamics of the A-site cation in 2D OIHP crystals. The study further unveiled that the dynamics of the A-site cation in 2D OIHP is affected by the crystal structures, which are influenced by the choice of the organic spacers. The interplay between the rigidity of the organic spacers and the A-site cations dynamics of 2D OIHPs is clearly unveiled, even though there is no direct bonding between the A-site cations and the organic spacers. Our results may provide a deep insight into the future design of 2D OIHPs at a molecular level.

## Methods

**Synthesis of 2D organic–inorganic hybrid perovskite**. The chemicals, including lead(II) oxide (PbO, ≥99.9%), 57% aqueous hydriodic acid (HI) in $H_2O$, 50% aqueous hypophosphorous acid ($H_3PO_2$) in $H_2O$, 99.5% 1-butylamine (BA), 99% 2-phenethylamine (PEA), 99% methyl-$d_3$-amine hydrochloride and ≥99% natural-abundance methylamine hydroiodide (MAI), were purchased from Sigma-Aldrich (St. Louis). The synthesis of $^{13}C$-methyl-$^{15}N$-ammonium iodide, $^{13}C,^{15}N$-MAI is described in the Supplementary Methods. The 2D OIHP crystals were synthesised according to Chen et al.[28]. Briefly, a mixed solution containing 10 ml of hydriodic acid, 57 wt.% in $H_2O$ and 1.7 mL of hypophosphorous acid solution, 50 wt.% in

$H_2O$, was prepared to dissolve 10 mmol PbO at 80 °C and stirred continuously at 600 rpm to obtain a yellow coloured $PbI_2$ solution. BAI was obtained by adding 10 mmol BA in 5 mL HI solution incubated in an ice bath. The BAI solution was slowly added into the $PbI_2$ solution at 80 °C to obtain orange-colour precipitates, and then the mixed solution was heated up to 100 °C to dissolve the reprecipitates. The solution was cooled down to room temperature to obtain orange flakes of 2D $(BA)_2PbI_4$ ($n = 1$) compound. For the synthesis of 2D $(BA)_2(MA)Pb_2I_7$ ($n = 2$), a stoichiometric quantity of the 5 mmol MAI was first dissolved into the $PbI_2$ solution to produce black precipitates of $MAPbI_3$, and then re-dissolved by heating up to 110 °C to obtain a clear yellow solution. BAI solution was obtained by adding 7 mmol BA in 5 mL HI solution incubated in an ice bath. The BAI solution was then added dropwise into the clear yellow solution at 110 °C. Finally, the solution was cooled down to room temperature to obtain red flakes of 2D $(BA)_2MAPb_2I_7$ ($n = 2$) compound. Similar procedures were followed for preparing 2D $(PEA)_2(MA)_{n-1}Pb_nI_{3n+1}$ crystals with $n = 1$ and $n = 2$, except that the different molar ratios of the precursors were used. The molar ratios of the precursors, PbO:MAI:PEA, were 1.72:0:3.45 and 6:18:1, for preparing the crystals with $n = 1$ and $n = 2$, respectively. Methyl-$d_3$-amine hydrochloride was used to replace the nature-abundance MA for the synthesis of 2D$(BA)_2(d_3$-MA)$Pb_2I_7$ ($n = 2$). $^{13}C$-methyl-$^{15}N$-ammonium iodide, abbreviated as $^{13}C,^{15}N$-MAI, was used to replace the natural-abundance MA for the synthesis of 2D $(BA)_2(^{13}C,^{15}N$-MA)$Pb_2I_7$ ($n = 2$) and $(PEA)_2(^{13}C,^{15}N$-MA)$Pb_2I_7$ ($n = 2$).

**Single crystals growth**. A saturated 2D OIHP solution was prepared by dissolving 2D OIHP compounds in a mixed solution containing 10 ml of hydriodic acid, 57 wt.% in $H_2O$, and 1.7 ml of hypophosphorous acid solution, 50 wt.% in $H_2O$, under stirring at a constant temperature of 60 °C in an oil bath. The well-stabilised 2D OIHP saturated solution was allowed to evaporate at a constant temperature of 62 °C for several days to obtain the high-quality 2D OIHP crystals. The crystals were carefully stored in a glove box to avoid moisture.

**Solid-state NMR experiments**. REDOR experiments were carried out on a wide-bore 14.1-T Bruker AVIII spectrometer equipped with a 3.2-mm triple-resonance magic-angle-spinning (MAS) probe head. The Larmor frequencies for $^1H$, $^{13}C$ and $^{15}N$ are 600.21, 150.92 and 60.81 MHz, respectively. The sample spinning rate was 10 kHz. The $^{13}C$ polarisation was built up by the cross-polarisation (CP) scheme with a $\pi/2$ pulse of 5 µs, and a CP contact of 1.5 ms. The $\pi$ pulse durations for $^{13}C$ and $^{15}N$ were 10 and 12 µs, respectively. During the REDOR sequence, the $^1H$ CW decoupling scheme with a 100-kHz rf field was adopted, while the $^1H$ TPPM decoupling with a 70 kHz rf field as used during signal acquisition. Static $^2H$ spectra were acquired on a wide-bore 11.7-T Bruker AVIII spectrometer equipped with a 3.2 mm MAS probehead. The Larmor frequency for $^2H$ is 76.75 MHz and the spectra were acquired by a solid echo sequence with two $\pi/2$ pulses of 6 µs and an interpulse delay of 50 µs, while the recycle delay was 1 s. $^1H$ CW decoupling of 50 kHz of field strength was applied during the signal acquisition.

## Data availability
The raw data collected in this study have been deposited in the figshare archive under accession code 19245843.v2.

## Code availability
The Mathematica codes used for the data fitting in Fig. 2b and for simulating the Supplementary Fig. 4 in this study have been deposited in the figshare archive under accession code 19245843.v2.

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

## Acknowledgements

This work is supported by Ministry of Science and Technology (MOST) (Taiwan), Academia Sinica iMate Program (AS-iMATE-111-33) and Career Development Award (AS-CDA-109-M03) for the funding. We thank Prof. Mark S. Conradi, Dr. Michitoshi Hayashi, Dr. Liang-Yan Hsu and Dr. Ching-Ming Wei for the valuable discussions and thank Mr. Boyan Andreychin for technical assistance.

## Author contributions

T.Y.Y. and C.W.C. conceived the idea and supervised this work; C.C.L. performed the synthesis of 2D OIHP crystals, PXRD, PL and the absorption experiments; C.Y.H. and Y.C.W. performed the synthesis of 3D MAPbI₃ crystals; T.Y.Y., S.J.H. and P.H.W. performed NMR experiments; T.P.C. performed temperature-dependent PXRD and DSC; T.P.C. and C.C.C. performed temperature-dependent PL; V.M.G., D.R. and O.P. performed the synthesis of $^{13}C$-methyl-$^{15}N$-ammonium iodide and discussed the results; T.Y.Y., C.W.C., P.T.C., R.S. and C.C.L. discussed and analysed the results; T.Y.Y., C.W.C., V.M.G. and C.C.L. wrote the manuscript together.

## Competing interests

The authors declare no competing interests.

**Additional information**

