## [Peer Review File · Nature Communications]

REVIEWER COMMENTS

Reviewer #1 (Remarks to the Author):

Nat Comm

Lin et al. investigate reorientational dynamics of methylammonium cations in 2D and 3D organic-inorganic halide perovskites (OIHP) using a specialized NMR probe (Rotational echo double resonance, REDOR) to interrogate the A site dynamics. Related solid-state NMR methods have seen increased use in studies directed towards understanding how A site dynamics influence optoelectronic properties. Issues mentioned with related techniques concern the ability to resolve these dynamics due to overlapping transitions for organic spacers between perovskite octahedra. They also use an isotopic labeling approach to better resolve A-site dynamics. While I do not completely understand the REDOR technique it does appear that they are capable of reliably discriminating between A-site and organic spacer contributions to the NMR signals. The work appears well executed but the paper in its current form is difficult to follow because and it is not intuitive for non-specialists. Furthermore, it is difficult to understand the significance of the results and how they pertain to particular perovskite structures. I believe the work is publishable but needs some revisions to improve the readability for readers to appreciate the results. Comments appear below.

Comments:

- It would be extremely helpful if the authors included a cartoon of the structures they are discussing and the proposed dynamics being measured from experiment. While many understand the general perovskite structure it is harder to envisage lower dimensional systems and how they differ from conventional systems in both structure and properties.

- p. 7 lines 200-205: They state that “reorientational motion of MA is absent.” in some systems but it is not clear why this is the case?

- Several groups have investigated the effect of electric fields on 2D OIHP electronic properties but it's not clear to me how the measured values of MA dipole reorientation compare to those reported here. It would help to have a better contextual link here since it is not clear to me if this is due to the experimental probe influence or a natural property. Additionally, there is little connection to why the results are significant for a particular application or class of material. For example, how does the interplay between A-site and spacers potentially impact optoelectronic properties of 2D OIHP.

- Fig. 4 caption states the contents contain PL spectra but they only show temperature dependent XRD and NMR. Am I missing something?

- p. 4 PEA was not defined earlier in the manuscript

- p. 5 line 134: should "sorely" be "solely"?

Reviewer #2 (Remarks to the Author):

:

Summary

The authors have studied the dynamics of the methylammonium ion in a number of perovskites, showing that the ion motion depends on nature of the solid phase and the constituents that make up the crystal. The NMR techniques employed required a lengthy synthetic effort. More sensitive results could have been obtained by using ^2H NMR as specified in the detailed report and ref 11 of the manuscript with use of a partially deuterated compound. The authors also neglected to use available crystallographic data to good advantage. A good visual representation would have helped the reader in understanding the various structural features discussed in the paper.

General comments going through the text

For obtaining information on the dynamics of the methylammonium ion the authors could have designed a far more effective approach. In Lines 93-107 the authors construct an argument to justify their experiments, saying that other techniques for studying the dynamics will fail because of spectral overlap. In fact the statement on lines 100-102 is nonsense – a ^2H NMR lineshape study would tell them far more than their results. They could have looked at CD_3NH_3^+ and CH_3ND_3^+ (by D_2O exchange) without all the effort and expense of producing $^{13}\text{C}/^{15}\text{N}$ doubly labelled CH_3NH_3^+ . Simple exchange with D_2O would also deuterate the NH_3 groups of the spacer ions, but since the spacers are likely less dynamic than the MA ion their ^2H lineshape will be considerably broader than that of MA. The authors the inadequacy of ^2H and ^{14}N relaxation studies, but such studies would not be necessary. ^2H NMR lineshapes as determined from quadrupolar interactions would give a

quantitative description of the motion of the C-N axis. The authors mention that the dynamics change, but do not provide a quantitative description.

Line 61: better to say “octahedral unit” rather than cage

Line 71 paragraph: It would be a great benefit to have a figure somewhere here showing a schematic structure of the 2D compounds, so the reader knows up-front what they are talking about.

Furthermore, the single crystal X-ray structure of 2D BA2MAPb2I7 (one of the compounds of this study) is known and should be referenced in this paragraph –yet this is their very last reference (45) ! – and they only reference it once (line 196) for the C-N distance for comparison with their REDOR results. This structural information is a very important reference and is needed for understanding this paper. Likewise, there is a crystal structure for their PEA compound which they have not referenced at all: in the supplemental data of JACS 113, 2328-2330 (1991) Calabrese et al. Both structures have problems with disorder, something that needs to be pointed out.

Line 111 and in many other places: what does the U- label stand for, and is it necessary ?

Line 116-118: there isn't any unequivocal revelation about the environment of MA from their results !

Line 151-155: more likely an impurity phase rather than a structural defect.

Line 163-167: - how do the results suggest this? – As much could be inferred from the crystal structures. The results may be consistent with the structural differences.

Line 196: first and only reference to the structural paper.

Line 202-203: Sentence “The faster the reorientational motion ..” It is not a question of faster and faster reorientation – though it just needs to be fast on the NMR timescale – as I understand it the REDOR fraction is related to the reduced effective dipolar coupling brought about by the reorientational averaging (a function of angular changes and populations). The authors have to distinguish between the rate and the geometry of the motion. Once the line shapes are in the fast motion limit the dipolar couplings do not change anymore. If changes with temperature are observed it will be due to changes in populations.

Lines 208-220, and lines 231-235: with reference to the 3D compound MAPbI3 they could compare the reduction in REDOR fraction (dipolar coupling) as temperature increases to the reduction in quadrupolar coupling for CD3NH3 and CH3ND3 obtained in the paper referenced as ref. 11).

Line 235: “motion of the MA did not change significantly in this temperature range” -- in fact the 2H results (ref.11) are much more sensitive and do show a small but gradually reducing quadrupole coupling constant CQ.

Lines 217-219: Again this suggests that the minor component is due to an impurity phase.

Line 227: as found in the crystal structure papers

Line 228-230: -- it does it by changing the crystal structure !

Line 243: `... is spacer dependent` -- again i.e. structure dependent.

Line 254: The authors should mention the DSC result here, as it provides confirmation of the phase change and its temperature.

Line 259: conformational change – maybe, but might also be doubling of content of asymmetric unit in the crystal

Line 270: How big is the shift?

Line 286-287: It isn't obvious how the author's results show that the structural deformation of the Pbl6 layers – and again around line 327 .. There is kind of a circular argument here --- the single crystal structures will show variations in the similar, but not identical, Pbl layers which affect the dynamics of the CH₃NH₃ ion enclosed within – so a change in MA dynamics indicates a change in its environment from one phase to another, but it does not tell you what that change in environment is.

Figure 1(b): there does not appear to be an `overlap problem` between the MA line and the organic spacer lines for the PEA compound ! – but, granted, the isotopic labelling does greatly enhance the intensity.

Figure 4: The caption mentions photoluminescence (PL) spectra but these are not in the figure – they have been moved to the supplemental section. I also note in Fig.4(d) that one of the ¹³C aromatic peaks of the PEA reduces quite dramatically as T increases – probably a sign of reorientation about the phenyl axis.

“XRD” results:

The authors do not explain their “XRD” patterns (which ought to be labelled throughout as PXRD)– the series of equally spaced peaks have been indexed and these reflections relate to the long axis of the unit cell. For both compounds they could at least give the derived axis parameter and compare it with the single crystal information as this would help confirm they have the right materials. They could also point out that these long axis parameters reflect the different spacings between the Pb₂I₇ layers (which they would have found out earlier if they had looked at the single crystal papers).

Compare crystal parameters:

(BA)₂MAPb₂I₇ (ref 45): At 293K, Orthorhombic Cc2m, a=8.9470, b=39.347, c=8.8589, Z=4

(PEA)₂MAPb₂I₇ (Calabrese): At 294K, Triclinic P-1, a=8.794, b=8.792, c=22.766, $\alpha=94.02^\circ$, $\beta=97.02^\circ$, $\gamma=90.18^\circ$, Z=2

From a rough measurement of the PXRD line spacings in figure S2 I calculate b=39.05 for the BA material and c=44.22 for the PEA material using their indexing numbers – note that the latter would correspond to a doubling of the unit cell given in the single crystal study (in line with the different Z values).

Supplementary data:

Fig.S1: The PXRD patterns are not exactly the same as those shown in fig.4 of the main text – so are these at a different temperature, perhaps room temperature ? Do they have an explanation for the different intensity patterns between the natural abundance and labelled samples ?

Fig. S3: For the BA compound going through the phase transition the shift increases by how much? What is the interpretation of the fairly substantial intensity decrease ?

Fig. S4: The DSC (is this warming or cooling ?) very clearly shows a phase transition only for the BA compound, onset ~280K. This ought to be stated directly in the main text.

Fig. S5 caption uses (c) and (d) instead of (a) and (b) as in the figure itself. The spectra are for the 2D compounds without MA , and for the BA compound in (a)they ought to point out that there are likely two phase transitions evident from the changes in the C3 and C4 peaks.

Fig. S6: The scale is barely legible. Assignments ?

Fig. S7 Presumably the splitting is from J-coupling to ^{15}N , if so point this out.

The work by Lin et al. is a study of methylammonium dynamics in Ruddlesden-Popper lead halides based on phenylethylammonium (PEA) and butylammonium (BA). The authors use uniformly ^{15}N and ^{13}C labelled MA to investigate the dipolar coupling strength between the ^{15}N - ^{13}C spin pair in three different materials and as a function of temperature. The authors record REDOR data which they use to conclude that the dynamics is qualitatively different in the PEA- and BA-based materials. The experimental data is of high quality. However, the analysis is relatively general and only focuses on qualitative differences between the MA dynamics in the different materials. That said, the authors have recorded quantitative REDOR data which should allow them to perform more in-depth analysis to obtain quantitative information on the spatial restriction and correlation time of the MA motion. This analysis will render the work substantially more prominent and of interest to the broad readership of *Nature Commun.* working in the fields of halide perovskite optoelectronics and solid-state NMR. There are also a number of highly relevant references which have been overlooked by the authors and should be acknowledged. This reviewer therefore recommends that the work should be considered for publication after the authors have addressed the following queries:

“This result indicates that the MA in 2D (PEA)₂([U-13C,U-15N]MA)Pb₂I₇ (n = 2) underwent a more restricted reorientational motion than the MA in 2D (BA)₂([U-13C,U-15N]MA)Pb₂I₇ (n = 2), suggesting that the local environments for MA in these two materials were quite different.”
“The above results suggest that the reorientational motion of the A-site cations due to cooling in 2D OIHP crystals is spacer-dependent.”

My main criticism regards the fact that the authors only draw qualitative conclusions regarding the MA dynamics (as illustrated by the above excerpts), despite having recorded high-quality data allowing them to perform quantitative analysis. The two key parameters characterizing dynamics are the order parameter and the correlation time, and the authors at present do not make an attempt to calculate and compare these parameters here for the three classes of materials under study.

In order for the dipolar coupling to be averaged out, the motion has to be isotropic (i.e. the order parameter has to vanish). When the authors refer to the motion as “more restricted” they imply that the order parameter is higher. Similarly, there is no attempt to calculate the correlation times. The discussion will benefit substantially from interpreting the results in the context of the model-free approach of Lipari and Szabo, see <https://pubs.acs.org/doi/10.1021/ja00381a009>. There are a number of experimental works using this framework for interpreting REDOR data (e.g. <https://link.springer.com/article/10.1007/s10858-013-9787-x>), which should greatly facilitate the task.

“Until now, only the dynamics of the organic spacers at the 2D OIHP crystals of (4NPEA)₂PbI₄ and (PEA)₂PbI₄ with n = 1, which consist of no A-site cations, have been analysed based on the temperature dependence of the ssNMR spectral line shapes “

The authors should reference and acknowledge a study from 1996 which studied the dynamics of PEA in the n=1 material:

<https://www.degruyter.com/document/doi/10.1515/zna-1996-0805/html>

as well as more recent works, which reports solid-state NMR studies of spacer dynamics in various 2D OIHPs:

<https://onlinelibrary.wiley.com/doi/full/10.1002/aenm.201900284> (n=1 and n=3 material based on FA)

<https://pubs.acs.org/doi/full/10.1021/acs.chemmater.0c03958> (n=1 materials for a wide range of different spacers)

Beyond dynamics, solid-state NMR studies of 2D OIHP have been rare so far with only a few reports so far. Acknowledging the above-mentioned works as well as the one below will provide a better representation of the current state of the art:

<https://pubs.acs.org/doi/abs/10.1021/acs.chemmater.0c04078>

In addition, please define “4NPEA”.

Regarding the assignment of the minor MA component in figure 1a: its chemical shift corresponds to unreacted MAI – could the author report the two compounds on the same scale and comment? This assignment seems more plausible than invoking structural defects. It is also consistent with the higher CP efficiency (MA in MAI is rigid).

Minor points:

Please explain the notation: “U-“ (for uniform labelling)

Grammar/typos:

line 134: “sorely”

line 135: “the growth of 2D OIHP crystals consisting of the conventional naturally abundant MA as the A-site cation, were also synthesised”

Response letter for “*Direct Investigation of the Reorientational Dynamics of A-site Cations in 2D Organic-Inorganic Hybrid Perovskite by Solid-State NMR*”

Previous Manuscript ID : NCOMMS-21-03696

Below, we would like to address each of the reviewer’s comments point-by-point. We prepared two versions of manuscripts, one clear version and the other with changes highlighted in red.

Reviewer#1:

Comments:

Lin *et al.* investigate reorientational dynamics of methylammonium cations in 2D and 3D organic-inorganic halide perovskites (OIHP) using a specialized NMR probe (Rotational echo double resonance, REDOR) to interrogate the A site dynamics. Related solid-state NMR methods have seen increased use in studies directed towards understanding how A site dynamics influence optoelectronic properties. Issues mentioned with related techniques concern the ability to resolve these dynamics due to overlapping transitions for organic spacers between perovskite octahedra. They also use an isotopic labeling approach to better resolve A-site dynamics. While I do not completely understand the REDOR technique it does appear that they are capable of reliably discriminating between A-site and organic spacer contributions to the NMR signals. The work appears well executed but the paper in its current form is difficult to follow because it is not intuitive for non-specialists. Furthermore, it is difficult to understand the significance of the results and how they pertain to particular perovskite structures. I believe the work is publishable but needs some revisions to improve the readability for readers to appreciate the results. Comments appear below.

Q1. It would be extremely helpful if the authors included a cartoon of the structures they are discussing and the proposed dynamics being measured from experiment. While many understand the general perovskite structure it is harder to envisage lower dimensional systems and how they differ from conventional systems in both structure and properties.

Answer: Thanks for the reviewer's valuable suggestion. We have followed the reviewer's suggestion to add cartoon illustrations of 2D OIHP structural models with A-site cation undergoing reorientational motion. The cartoon illustration should be helpful for the readers to understand the structural differences between 2D layer perovskite and the conventional 3D perovskite.

Revision made: Fig. 1 has been modified in the revised manuscript.

Fig. 1 ^{13}C NMR characterisation spectra of 2D OIHP crystals ($n = 2$). (a) The structural models of 2D $(\text{BA})_2(\text{MA})\text{Pb}_2\text{I}_7$ ($n = 2$) and 2D $(\text{PEA})_2(\text{MA})\text{Pb}_2\text{I}_7$ ($n = 2$). (b) The ^{13}C CPMAS spectra of 2D $(\text{BA})_2(\text{MA})\text{Pb}_2\text{I}_7$ ($n = 2$) and 2D $(\text{PEA})_2(\text{MA})\text{Pb}_2\text{I}_7$ ($n = 2$) synthesised with $^{13}\text{C}, ^{15}\text{N}$ -MA, respectively. The spectra with 20x magnification are shown on the top, overlaid with the spectra of the materials synthesised with natural abundance MA. The two sets of spectra have been normalised by the height of the C1 carbon peak of BA and the aromatic carbon peak at 130.7 ppm, respectively. (c)

2D DCP MAS (^{13}C , ^{15}N) correlation spectra of 2D $(\text{BA})_2(^{13}\text{C}, ^{15}\text{N}\text{-MA})\text{Pb}_2\text{I}_7$ ($n = 2$) and 2D $(\text{PEA})_2(^{13}\text{C}, ^{15}\text{N}\text{-MA})\text{Pb}_2\text{I}_7$ ($n = 2$), respectively. The ^{13}C direct excitation MAS NMR spectra are overlaid on the top.

Q2. p. 7 lines 200-205: They state that “reorientational motion of MA is absent.” in some systems but it is not clear why this is the case?

Answer: Thanks for pointing out that our description was not clear. Here, we used a ^{13}C , ^{15}N -MAI crystal powder sample as a reference, where the MA cations are fixed in the crystal lattice. In contrast, the MA cations in the PbI_6 octahedral unit of 2D and 3D crystals have some degree of freedom for reorientational motion. We have revised our manuscript to make the description clear to the reader.

Revision made: “A powder sample of ^{13}C , ^{15}N -MAI crystals was used as a reference, where the reorientational motion of the ^{13}C - ^{15}N vector is absent. Fitting the experimental $^{13}\text{C}\{^{15}\text{N}\}$ REDOR results (orange open circles) to a simulated dephasing curve (orange dashed line) indicate a C-N bond length of 1.51 Å, consistent with published single crystal X-ray crystallography bond lengths of 1.469 to 1.516 Å^{32,47-50}.”

Q3. Several groups have investigated the effect of electric fields on 2D OIHP electronic properties but it’s not clear to me how the measured values of MA dipole reorientation compare to those reported here. It would help to have a better contextual link here since it is not clear to me if this is due to the experimental probe influence or a natural property. Additionally, there is little connection to why the results are significant for a particular application or class of material. For example, how does the interplay between A-site and spacers potentially impact optoelectronic properties of 2D OIHP.

Answer: We have included additional results from photoluminescence (PL) spectroscopy characterization of 2D OIHPs. Our results demonstrate that the choice of the spacer molecule in 2D OIHPs affects the optoelectronic properties of the material.

Revision made: We include the PL spectra of the 2D OIHPs in Fig. S7. We showed that the optoelectronic property of 2D $(\text{BA})_2\text{MAPb}_2\text{I}_7$ ($n = 2$), in terms of the peak-position shift of the PL spectra, in response to cooling. In contrast, no peak-position shift of the PL spectra of 2D $(\text{PEA})_2\text{MAPb}_2\text{I}_7$ ($n = 2$) was observed in the cooling

process. Our results indicated the connection between the choice of the spacer molecule in 2D OIHPs affects the dynamics of A-site cations and the optoelectronic property of the material. Accordingly, the following description was added in the revised manuscript.

“It is well-known that the structural deformation of the octahedral layers, induced by changing the packing geometry of the organic spacers, may strongly affect the optical and electronic properties of 2D OIHPs^{32,38,40}. Photoluminescence spectroscopy (PL) during the cooling from 300 to 250K revealed a significant blue shift from 581 to 574 nm in 2D (BA)₂(¹³C,¹⁵N-MA)Pb₂I₇ (n = 2), as shown in Fig. S7. The corresponding PL emission peak of 2D (PEA)₂(¹³C,¹⁵N-MA)Pb₂I₇ (n = 2) remained unshifted. The results suggest that the phase change or structural deformation of 2D OIHPs induced by changing the packing geometry of the organic spacers, may strongly affect the optical properties of 2D OIHPs.”

Q4. Fig. 4 caption states the contents contain PL spectra but they only show temperature-dependent PXRD and NMR. Am I missing something?

Answer: Thanks to the reviewer for pointing out the mistake. Fig. 4 in the previous version is now Fig. 5 in the revised version.

Revision made: We have revised the caption of Fig. 5 and included the PL spectra in Fig. S7.

Q5. p.4 PEA was not defined earlier in the manuscript.

Answer: We have defined the PEA in introduction. This sentence “Ruddlesden-Popper perovskites are a typical example of layered 2D organic-inorganic hybrid perovskites having the generic chemical formula A'₂A_{n-1}M_nX_{3n+1}. In this formula, A' represents an organic spacer, such as long chain alkylammonium cation (e.g. 1-butylammonium, BA⁺) or phenyl alkylammonium cation (e.g. 2-phenethylammonium, PEA⁺), A is an organic cation, M is a metal, X is a halide, and n is the number of octahedral slabs per unit cell²⁵⁻²⁸.” has defined PEA already.

Q6. - p. 5 line 134: should “sorely” be “solely”?

Answer: Thanks for the reviewer’s correction.

Revision made: The incorrect word “sorely” in line 134 has been replaced with the correct word “solely” in the revised manuscript.

Reviewer#2:

Comments:

For obtaining information on the dynamics of the methylammonium ion the authors could have designed a far more effective approach. In Lines 93-107 the authors construct an argument to justify their experiments, saying that other techniques for studying the dynamics will fail because of spectral overlap. In fact, the statement on lines 100-102 is nonsense a ^2H NMR lineshape study would tell them far more than their results. They could have looked at CD_3NH_3^+ and CH_3ND_3^+ (by D_2O exchange) without all the effort and expense of producing $^{13}\text{C}/^{15}\text{N}$ doubly labelled CH_3NH_3^+ . Simple exchange with D_2O would also deuterate the NH_3 groups of the spacer ions, but since the spacers are likely less dynamic than the MA ion their ^2H lineshape will be considerably broader than that of MA. The authors the inadequacy of ^2H and ^{14}N relaxation studies, but such studies would not be necessary. ^2H NMR lineshapes as determined from quadrupolar interactions would give a quantitative description of the motion of the C-N axis. The authors mention that the dynamics change, but do not provide a quantitative description.

Answer: We would like to thank the reviewer for the valuable comments and suggestions. REDOR NMR measures the dipolar coupling between the ^{13}C and ^{15}N nuclei of MA, which is modulated by the molecular reorientational motion of MA. Thus, we claimed that REDOR NMR measurement directly reflects this motional averaging process. We agree that the same molecular motion also averages the deuterium quadrupole coupling and, as a result, ^2H NMR lineshape also reflects the motion of the ^2H spin. In the case of using CD_3NH_3^+ to replace the natural abundance MA in the synthesis of 2D OIHPs, the quadrupole splitting (ν_Q), measured cusp to cusp of the ^2H spectrum, would be reduced from the static value of ~ 120 kHz to ~ 40 kHz in the presence of rapid C3 rotation, and the existence of additional reorientational motion of the C3 axis, the C-N vector in this case, would result in narrower spectral lineshape.

Thus, even though it is not a direct observation of the motion of the C-N vector, ^2H NMR lineshape does allow one to learn valuable information about the dynamics of the C-N vector. In this regard, we thank the reviewer's suggestion and happily performed additional experiments to record ^2H NMR spectra of 2D $(\text{BA})_2(\text{d}_3\text{-MA})\text{Pb}_2\text{I}_7$ ($n = 2$) and 2D $(\text{PEA})_2(\text{d}_3\text{-MA})\text{Pb}_2\text{I}_7$ ($n = 2$) in our study, Fig. 4, where CD_3NH_3^+ was used to replace the natural abundance MA in the synthesis of 2D OIHPs. We indeed learned quite a lot of valuable information from the additional experimental results and found that together with REDOR NMR and ^2H NMR, we can get more complete dynamics information of MA in 2D OIHPs. We have included the most valuable additional information learned from ^2H NMR lineshapes of 2D OIHPs is the existence of multiple motional modes of MA. At the same time, we would like to point out that together with the usage of $^{13}\text{C}, ^{15}\text{N}$ -MA, REDOR NMR, being performed under MAS, allowed us to reveal the existence of the minor MA component in 2D $(\text{BA})_2(^{13}\text{C}, ^{15}\text{N}\text{-MA})\text{Pb}_2\text{I}_7$ ($n = 2$) and to study its dynamics. In addition, we observed a 4% change of $^{13}\text{C}\{^{15}\text{N}\}\text{REDOR}(\Delta S/S_0)_{2.4\text{ms}}$ value of 2D $(\text{PEA})_2(^{13}\text{C}, ^{15}\text{N}\text{-MA})\text{Pb}_2\text{I}_7$ ($n = 2$), as shown in Fig. 3, when the temperature was cooled from 298 to 243K, while very little ^2H NMR lineshape change of 2D $(\text{PEA})_2(\text{d}_3\text{-MA})\text{Pb}_2\text{I}_7$ ($n = 2$) was observed in the same temperature range, as shown Fig. 4. Finally, we agree that the isotopically labelling material $^{13}\text{C}, ^{15}\text{N}$ -MA is not cheap. Thus, in this manuscript we would like to share with the community an inexpensive synthesis method for producing it.

Revision made: We thank the reviewer for reviewing our manuscript carefully and providing many valuable comments and suggestions. We took the reviewer's suggestion to include additional ^2H NMR spectra of 2D OIHPs prepared with CD_3NH_3^+ , as shown in Fig. 4. Accordingly, we have included the description of the line shape analysis of the ^2H NMR spectra in the Results section, as well as in the Discussions section.

In addition, we have also revised the abstract as follows.

“Limited methods are available for investigating the reorientational dynamics of A-site cations in two-dimensional organic-inorganic hybrid perovskites (2D OIHPs), which play a pivotal role in determining their physical properties. Here we describe a novel approach to study the dynamics of A-site cations using solid state NMR and stable isotope labelling. ^2H NMR of 2D OIHPs incorporating methyl-d₃-ammonium cations (d₃-MA) revealed the existence of multiple modes of reorientational motions of MA.

Rotational-echo double resonance (REDOR) NMR of 2D OIHPs incorporating ^{15}N - and ^{13}C -labeled methyl ammonium cations (^{13}C , ^{15}N -MA) reflected the averaged dipolar coupling between the C and N nuclei undergoing different modes of motions. Our study revealed the interplay between the A-site cation dynamics and the structural rigidity of the organic spacers, so providing a molecular-level insight into the design of 2D OIHPs.”

Fig. 4 ^2H NMR spectra of 2D OIHP crystals ($n = 2$). The full-scaled ^2H spectra of (a) $2\text{D}(\text{BA})_2(\text{d}_3\text{-MA})\text{Pb}_2\text{I}_7$ ($n = 2$) and (b) $2\text{D}(\text{PEA})_2(\text{d}_3\text{-MA})\text{Pb}_2\text{I}_7$ ($n = 2$) recorded at various temperatures, ranging from 298 to 243K.

Q1. Line 61: better to say “octahedral unit” rather than cage.

Answer: Thanks for the reviewer’s suggestion. Both the descriptions of “the PbI_6 octahedral unit” and “the PbI_6 octahedral cage” are frequently used in literature. (*APL Mater* **8**, 041104 (2020) and *Nat. Commun* **8**, 15688 (2017)) We have happily followed the reviewer’s suggestion to change the description of “the PbI_6 octahedral cage” to “the PbI_6 octahedral unit”.

Revision made: The description of “the PbI_6 octahedral unit” is now used in the revised manuscript.

Q2. Line 71 paragraph: It would be a great benefit to have a figure somewhere here showing a schematic structure of the 2D compounds, so the reader knows up-front what they are talking about. Furthermore, the single crystal X-ray structure of 2D BA₂MAPb₂I₇ (one of the compounds of this study) is known and should be referenced in this paragraph –yet this is their very last reference (45)! – and they only reference it once (line 196) for the C-N distance for comparison with their REDOR results. This structural information is a very important reference and is needed for understanding this paper. Likewise, there is a crystal structure for their PEA compound which they have not referenced at all: in the supplemental data of JACS 113, 2328-2330 (1991) Calabrese et al. Both structures have problems with disorder, something that needs to be pointed out.

Answer: We answer this question together with the Q7 raised by the reviewer #2. Thanks for the reviewer’s valuable suggestion. We have followed the reviewer’s suggestion to add cartoon illustrations of 2D OIHP structural models with A-site cation undergoing reorientational motion. The cartoon illustration should be helpful for the readers to understand the structural differences of 2D layer perovskite and the conventional 3D perovskite.

We also agree with the suggestion to cite the papers of the single crystal X-ray structure of 2D (BA)₂MAPb₂I₇ and (PEA)₂MAPb₂I₇ in the manuscript. These include references 32, 47-50 in the revised manuscript, and the more recent works (*Chem. Mater* **28**, 2852-2867 (2016) and *J. Am. Chem. Soc* **113**, 2330-2332 (1991)). These studies indicate the existence of the structural disorder of the interlayer cations and the A-site cations. The newly included ²H NMR results echo these findings. Finally, we have included all the recent references reporting the bond length of the C-N vector and discussed them in the revised manuscript, as shown in the table below.

Revision made: Fig. 1 and references have been modified in the revised manuscript.

Table R1. The bond length of the C-N vector in 2D OIHP from recent references.

Materials	C-N vector length (Å)	Literature	Temperature
(BA) ₂ MAPb ₂ I ₇	1.495	APL Mater. 6, 114207 (2018)	225K
(BA) ₂ MAPb ₂ I ₇	1.512	Chem. Mater. 28, 2852-2867 (2016)	293K
(BA) ₂ MAPb ₂ I ₇	1.504	Chem. Mater. 31, 5592-5607 (2019)	250K
(BA) ₂ MAPb ₂ I ₇	1.498	Chem. Mater. 31, 5592-5607 (2019)	300k
(PEA) ₂ MAPb ₂ I ₇	1.469	J. Am. Chem. Soc. 113, 2328-2330 (1991)	294K
(PEA) ₂ MAPb ₂ I ₇	1.516	J. Phys. Chem. Lett. 11, 6551-6559 (2020)	100K

Q3. Line 111 and in many other places: what does the U- label stand for, and is it necessary?

Answer: We answer this question together with the Q6 raised by the reviewer #3. [U-¹³C, U-¹⁵N] MA stands for uniformly ¹³C and ¹⁵N labeled MA. To ease the reading, the ¹³C- and ¹⁵N-enriched methylammonium is abbreviated as ¹³C,¹⁵N-MA in the revised manuscript.

Revision made: The ¹³C- and ¹⁵N-enriched methylammonium is abbreviated as ¹³C,¹⁵N-MA in the revised manuscript.

Q4. Line 116-118: there isn't any unequivocal revelation about the environment of MA from their results!

Answer: We would like to show that the two environments of MA were clearly observed in 2D ¹³C-¹⁵N correlation DCP MAS spectrum of 2D (BA)₂(¹³C,¹⁵N-MA)Pb₂I₇ (n = 2). In our case, only ¹³C,¹⁵N-MA contributes to the 2D ¹³C-¹⁵N correlation DCP MAS NMR signal. Thus, the two chemical environments of MA were clearly revealed. Moreover, the peak position of ¹³C,¹⁵N-MA in the PEA-based 2D OIHP is different from the positions of the two MA components in the BA-based 2D OIHP, showing that the chemical environments are different.

Revision made: The last paragraph of the introduction has been revised as follows.

“To study the dynamics of MA, we first applied two-dimensional ¹³C-¹⁵N correlation double cross-polarization magic-angle spinning spectroscopy⁴⁴, 2D (¹³C,¹⁵N) DCP MAS, to observe the NMR signal of MA with a substantial sensitivity enhancement and without the interference of the signal of the spacer cations. REDOR NMR and ²H NMR were further employed to study the reorientational dynamics of MA in 2D OIHPs, which provides motional average to the dipolar coupling between the ¹³C and ¹⁵N nuclei of ¹³C,¹⁵N-MA and to the deuterium quadrupole coupling of CD₃NH₃⁺, respectively. Accordingly, both the environments and the dynamics of A-site molecular cations can be revealed. We further characterized the structural and optoelectronic properties of 2D OIHPs using powder X-ray spectroscopy (PXRD), absorption spectroscopy and photoluminescence spectroscopy (PL), in addition to ¹³C CPMAS NMR characterization of the incorporated spacer molecules at the various temperatures. Our

PXRD and PL results indicated that the choice of the spacer could influence the structures and the optoelectronic properties of 2D OIHPs, as mentioned in the literature^{32,38,40}. We further showed that a 1-butylammonium spacer (BA^+) is less rigid than a 2-phenethylammonium spacer (PEA^+) according to our ^{13}C CPMAS NMR results. Finally, the detection of the reorientational dynamics of MA by the two ssNMR methods reported here allows us to further examine the interplay between the rigidity of organic spacers and the dynamics of the A-site cations in 2D OIHPs. The present study complements previous NMR studies focusing on the spacer molecules or the frameworks^{28,40-43}, and should provide new insights into the future design of 2D OIHP materials.”

Q5. Line 151-155: more likely an impurity phase rather than a structural defect.

Answer: We answer this question together with the Q11 raised by the reviewer #2 and the Q5 raised by the reviewer #3. We have plotted the ^{13}C NMR spectra of 2D $(\text{BA})_2(^{13}\text{C},^{15}\text{N-MA})\text{Pb}_2\text{I}_7$ ($n = 2$) and the precursor $^{13}\text{C},^{15}\text{N-MAI}$ crystal powder on the same scale, as shown in Fig. S3. The possibility of the minor MA component being the unreacted $^{13}\text{C},^{15}\text{N-MAI}$ has been ruled out based on the difference observed in their ^{13}C chemical shifts. In addition, the mixed phase of 2D OIHPs can easily be verified by powder X-ray diffraction (PXRD) characterization and absorption spectroscopy. (*Nat. Nanotechnol* **15**, 969 (2020)). Based on our results shown in Fig. S1, there are only one series of periodic repetitions of Miller planes without any additional PXRD patterns in both 2D $(\text{BA})_2\text{MAPb}_2\text{I}_7$ ($n = 2$) and the one prepared with $^{13}\text{C},^{15}\text{N-MA}$, 2D $(\text{BA})_2(^{13}\text{C},^{15}\text{N-MA})\text{Pb}_2\text{I}_7$ ($n = 2$). Moreover, only one strong excitonic absorption peak can be observed in the absorption spectra of both 2D $(\text{BA})_2(\text{MA})\text{Pb}_2\text{I}_7$ ($n = 2$) and 2D $(\text{BA})_2(^{13}\text{C},^{15}\text{N-MA})\text{Pb}_2\text{I}_7$ ($n = 2$) as shown in Fig. S2. Therefore, the minor component (27.2 ppm peak) might more likely be structural impurity (eg. one PbI_6 octahedral unit missing or one spacer missing) according to recent work (*Nat. Nanotechnol* **15**, 969 (2020)). The amount of the minor MA component was estimated to be only 3% of the total amount of MA and its existence does not influence the conclusion of this work.

Revision made: The following description as well as Fig. S3 were added to the revised manuscript.

“Comparison of the ^{13}C NMR spectra of the $^{13}\text{C},^{15}\text{N}$ -perovskite and the $^{13}\text{C},^{15}\text{N}$ -methylammonium iodide precursor (Fig. S3) ruled out the possibility of the minor MA component being the unreacted $^{13}\text{C},^{15}\text{N}$ -methylammonium iodide precursor. Moreover, the PXRD and UV data (Fig. S1 and S2) showed one series of periodic repetitions of Miller planes and one excitonic absorption peak in both the $^{13}\text{C},^{15}\text{N}$ -labeled and the unlabeled 2D $(\text{BA})_2\text{MAPb}_2\text{I}_7$ ($n = 2$). Thus, the minor MA spectral component indicates the presence of a minor structural impurity, as suggested in a recent work⁴⁵.”

Fig. S3 The ^{13}C CPMAS NMR spectra of $^{13}\text{C},^{15}\text{N}$ -MAI crystal powder (top) and 2D $(\text{BA})_2(^{13}\text{C},^{15}\text{N}\text{-MA})\text{Pb}_2\text{I}_7$ ($n = 2$) (down), respectively. The ^{13}C resonance peak of MAI is marked with the red dashed line to aid the comparison of the chemical shifts of the MAI crystal powder and the minor MA component.

Q6. Line 163-167: - how do the results suggest this? – As much could be inferred from the crystal structures. The results may be consistent with the structural differences.

Answer: Thanks for the reviewer’s comment. In general, the choice of the incorporated spacer cation in 2D OIHP is play an important role to design their structures and optoelectronic properties. (*Chem. Rev.* **121**, 2230-2291 (2021)) It has been reported that the manipulations of organic spacers may cause the structural rearrangement, or deformation, of inorganic perovskite unit in 2D OIHPs (*J. Phys. Chem. Lett* **10**, 2924-2930 (2019) and *Chem. Mater* **31**, 5592-5607 (2019)), which further influence their electronic band structures near the band edges, and the corresponding optical and

electronic behaviors (*Nat. Mater* **17**, 550-556 (2018) and *J. Am. Chem. Soc* **141**, 4521-4525 (2019)). Except for structure and optoelectronic properties, our results indicate that the dynamics and local environments of A-site cations in 2D OIHP are different between 2D OIHP with BA and PEA. According to theoretical work (*Nat. Commun* **6**, 7026 (2015)), the dynamics of A-site cations may significantly affect the optoelectronic properties and even the superior conversion efficiency of perovskite photovoltaics. Therefore, our study suggests that the choice of the organic spacer in 2D OIHP is a critical issue for better optoelectronic properties.

Revision made: The sentences between lines 163-167 in the original manuscript have been revised as follows.

“In summary, different local environments and the dynamics of the A-site cations were observed in 2D OIHP crystals incorporated with different organic spacers.”

Q7. Line 196: first and only reference to the structural paper.

Answer: We answer this question together with the Q2 raised by the reviewer #2. Briefly, we have followed the suggestion to cite the papers of the single crystal X-ray structure of 2D (BA)₂MAPb₂I₇ and (PEA)₂MAPb₂I₇ in the manuscript. These include references 32, 47-50 in the revised manuscript, and the more recent works (*Chem. Mater.* **28**, 2852-2867 (2016) and *J. Am. Chem. Soc.* **113**, 2330-2332 (1991)). These studies indicate the existence of the structural disorder of the interlayer cations and the A-site cations. The newly included ²H NMR results echo these findings. Finally, we have included all the recent references reporting the bond length of the C-N vector and discussed them in the revised manuscript, as shown in the table below.

Revision made: The revision has been described in response to the Q2 raised by the reviewer.

Table R1. The bond length of the C-N vector in 2D OIHP from recent references.

Materials	C-N vector length (Å)	Literature	Temperature
(BA) ₂ MAPb ₂ I ₇	1.495	APL Mater. 6, 114207 (2018)	225K
(BA) ₂ MAPb ₂ I ₇	1.512	Chem. Mater. 28, 2852-2867 (2016)	293k
(BA) ₂ MAPb ₂ I ₇	1.504	Chem. Mater. 31, 5592-5607 (2019)	250K
(BA) ₂ MAPb ₂ I ₇	1.498	Chem. Mater. 31, 5592-5607 (2019)	300k
(PEA) ₂ MAPb ₂ I ₇	1.469	J. Am. Chem. Soc. 113, 2328-2330 (1991)	294K
(PEA) ₂ MAPb ₂ I ₇	1.516	J. Phys. Chem. Lett. 11, 6551-6559 (2020)	100K

Q8. Line 202-203: Sentence “The faster the reorientational motion..” It is not a question of faster and faster reorientation – though it just needs to be fast on the NMR timescale – as I understand it the REDOR fraction is related to the reduced effective dipolar coupling brought about by the reorientational averaging (a function of angular changes and populations). The authors have to distinguish between the rate and the geometry of the motion. Once the line shapes are in the fast motion limit the dipolar couplings do not change anymore. If changes with temperature are observed it will be due to changes in populations.

Answer: We thank the reviewer’s comment and answer this question together with the Q1 raised by the reviewer #3. We took the suggestion to record the ^2H NMR spectra of $2\text{D}(\text{BA})_2(\text{d}_3\text{-MA})\text{Pb}_2\text{I}_7$ ($n = 2$) and $2\text{D}(\text{PEA})_2(\text{d}_3\text{-MA})\text{Pb}_2\text{I}_7$ ($n = 2$). The reorientational motion of MA provides motional averaging to the deuterium quadrupolar coupling of CD_3NH_3^+ , as well as to the dipolar coupling between the ^{13}C and ^{15}N spins of ^{13}C , ^{15}N -MA. Thus, we should be able to extract the information of the reorientational motion from both the ^2H NMR lineshape analysis and REDOR NMR. Our results indicated that these are the two complementary methods. We observed the multiple double-horned quadrupolar splitting patterns in the ^2H NMR spectra, suggesting the existence of multiple motional modes of MA incorporated in 2D OIHPs. On the other hand, REDOR NMR, exhibiting the high resolution power of MAS, revealed the existence of the 3% of the minor MA component in $2\text{D}(\text{BA})_2(^{13}\text{C}, ^{15}\text{N}\text{-MA})\text{Pb}_2\text{I}_7$ ($n = 2$).

Ideally, one should be able to construct a model to analyze both ^2H NMR line shapes and REDOR dephasing curve to extract the motional parameters quantitatively, provided the distribution of the C-N vector of MA relative to the PbI_6 octahedral unit is known. The different orientation preferences of MA may be associated with the different motional modes. However, unambiguous information is not available, preventing us from doing reliable analyses to extract the motional correlation times. The REDOR dephasing data at 2.4 ms can be correlated with the order parameters averaging over the different motional modes of the incorporated MA. The dynamics of the incorporated MA in BA- and PEA-based 2D OIHPs respond very differently to the temperature change. This main discovery is sufficiently supported with the provided evidence. We also like to thank the reviewer for pointing out some of the key references. They are all properly cited in the revised manuscript.

Revision made: We have included the suggested key references in the revised manuscript. These include the two references of the model-free approach^{53,54}, the REDOR derived order parameter, the previous studies of the spacer dynamics^{28,41,42} and relevant solid-state NMR works⁴³, listed below.

In addition to citing more references, we included the theoretical description of REDOR NMR in the presence of molecular motion to establish the relationship between the REDOR dephasing value and the order parameter, characterizing the reorientational motion. Moreover, we included the ²H NMR experiments kindly suggested by the reviewer #2 for comparison. These changes are highlighted in our revised manuscript.

Reference

53 Lipari, G. & Szabo, A. Model-Free Approach to the Interpretation of Nuclear Magnetic Resonance Relaxation in Macromolecules. 1. Theory and Range of Validity. *J. Am. Chem. Soc* **104**, 4546-4559 (1982).

54 Haller, J. D. & Schanda, P. Amplitudes and Time Scales of Picosecond-to-Microsecond Motion in Proteins Studied by Solid-State NMR: A Critical Evaluation of Experimental Approaches and Application to Crystalline Ubiquitin. *J. Biomol. NMR* **57**, 263-280 (2013).

28 Milic, J. V. *et al.* Supramolecular Engineering for Formamidinium-Based Layered 2D Perovskite Solar Cells: Structural Complexity and Dynamics Revealed by Solid-State NMR Spectroscopy. *Adv. Energy Mater* **9**, 1900284 (2019).

41 Ueda, T., Shimizu, k., Ohki, H. & Okuda, T. ¹³C CP/MAS NMR Study of the Layered Compounds [C₆H₅CH₂CH₂NH₃]₂[CH₃NH₃]_{n-1}Pb_nI_{3n+1} (n = 1, 2). *Z. Naturforsch. A* **51**, 910-914 (1996).

42 Dahlman, C. J. *et al.* Dynamic Motion of Organic Spacer Cations in Ruddlesden–Popper Lead Iodide Perovskites Probed by Solid-State NMR Spectroscopy. *Chem. Mater* **33**, 642-656 (2021).

43 Lee, J., Lee, W., Kang, K., Lee, T. & Lee, S. K. Layer-by-Layer Structural Identification of 2D Ruddlesden–Popper Hybrid Lead Iodide Perovskites by Solid-State NMR Spectroscopy. *Chem. Mater* **33**, 370-377 (2021).

Q9. Lines 208-220, and lines 231-235: with reference to the 3D compound MAPbI₃ they could compare the reduction in REDOR fraction (dipolar coupling) as temperature

increases to the reduction in quadrupolar coupling for CD_3NH_3 and CH_3ND_3 obtained in the paper referenced as ref. 11).

Answer: We would like to express our gratitude to the reviewer for making valuable suggestions to include ^2H NMR experiments. As mentioned in our response to the major comment raised by the reviewer, we agree that the molecular reorientational motion also averages the deuterium quadrupole coupling and, as a result, ^2H NMR lineshape also reflects the motion of the ^2H spin. In the case of using CD_3NH_3^+ to replace the natural abundance MA in the synthesis of 2D OIHPs, the quadrupole splitting (ν_Q), measured cusp to cusp of the ^2H spectrum, would be reduced from the static value of ~ 120 kHz to ~ 40 kHz in the presence of rapid C3 rotation, and the existence of additional reorientational motion of the C3 axis, the C-N vector in this case, would result in narrower spectral lineshape. Thus, even though it is not a direct observation of the motion of the C-N vector, ^2H NMR lineshape does allow one to learn valuable information about the dynamics of the C-N vector. In this regard, we thank the reviewer's suggestion and happily performed additional experiments to record ^2H NMR spectra of 2D $(\text{BA})_2(\text{d}_3\text{-MA})\text{Pb}_2\text{I}_7$ ($n = 2$) and 2D $(\text{PEA})_2(\text{d}_3\text{-MA})\text{Pb}_2\text{I}_7$ ($n = 2$) in our study, Fig. 4, where CD_3NH_3^+ was used to replace the natural abundance MA in the synthesis of 2D OIHPs. We indeed learned quite a lot of valuable information from the additional experimental results and found that together with REDOR NMR and ^2H NMR, we can get more complete dynamics information of MA in 2D OIHPs. We have included the most valuable additional information learned from ^2H NMR lineshapes of 2D OIHPs, which is the existence of multiple motional modes of MA.

Revision made: We took the reviewer's suggestion to include ^2H NMR spectra of 2D OIHPs prepared with CD_3NH_3^+ . Accordingly, the last paragraph of the introduction has been revised as described in our response to the major comment of the reviewer. In addition, we included the description of the ^2H NMR results in the experimental section and discussed the results in the discussion section accordingly.

Q10. Line 235: "motion of the MA did not change significantly in this temperature range" -- in fact the ^2H results (ref.11) are much more sensitive and do show a small but gradually reducing quadrupole coupling constant CQ.

Answer: Thanks for the reviewer's comment. We carefully examined the data reported in reference 11, where we also observed minor ^2H NMR lineshape change of $\text{CH}_3\text{ND}_3\text{PbI}_3$ in the temperature range of 294 to 237 K, which is comparable with the temperature range used in our experiment from 308 to 243K. Moreover, we have included ^2H NMR lineshape studies (Fig. 4) of both 2D $(\text{BA})_2(\text{d}_3\text{-MA})\text{Pb}_2\text{I}_7$ ($n = 2$) and 2D $(\text{PEA})_2(\text{d}_3\text{-MA})\text{Pb}_2\text{I}_7$ ($n = 2$). We have discussed the comparison of these two ssNMR methods in our response to the major comment. Briefly, according to our results, these are two complementary ssNMR methods. In the case of 2D $(\text{PEA})_2(\text{d}_3\text{-MA})\text{Pb}_2\text{I}_7$ ($n = 2$), very little ^2H NMR lineshape change was observed when the temperature was cooled from 298 to 243K (Fig. 4), while 4% change of $^{13}\text{C}\{^{15}\text{N}\}\text{REDOR}(\Delta S/S_0)_{2.4\text{ms}}$ value of 2D $(\text{PEA})_2(\text{d}_3\text{-MA})\text{Pb}_2\text{I}_7$ ($n = 2$) was measured during the cooling from 298 to 243K, as shown in Fig. 3. According to our results, the ^2H NMR lineshape analysis uniquely revealed the existence of the multiple motional modes. On the other hand, REDOR NMR measurements reflected the change of the averaged MA dynamics in response to the temperature change.

Revision made: The revision has been described in the response to the major comment of the reviewer, we have included the additional results of ^2H NMR line shape analysis and discussed them accordingly in the revised manuscript.

Q11. Lines 217-219: Again this suggests that the minor component is due to an impurity phase.

Answer: We have answered this question together with the Q5 raised by the reviewer #2 and the Q5 raised by the reviewer #3. We have plotted the ^{13}C NMR spectra of 2D $(\text{BA})_2(^{13}\text{C}, ^{15}\text{N}\text{-MA})\text{Pb}_2\text{I}_7$ ($n = 2$) and the precursor $^{13}\text{C}, ^{15}\text{N}\text{-MAI}$ crystal powder on the same scale, as shown in Fig. S3. The possibility of the minor MA component being the unreacted $^{13}\text{C}, ^{15}\text{N}\text{-MAI}$ has been ruled out based on the difference observed in their ^{13}C chemical shifts. In addition, the mixed phase of 2D OIHPs can easily be verified by powder X-ray diffraction (PXRD) characterization and absorption spectroscopy. (*Nat. Nanotechnol* **15**, 969 (2020)). Based on our results shown in Fig. S1, there are only one series of periodic repetitions of Miller planes without any additional PXRD patterns in both 2D $(\text{BA})_2\text{MAPb}_2\text{I}_7$ ($n = 2$) and the one prepared with $^{13}\text{C}, ^{15}\text{N}\text{-MA}$, 2D $(\text{BA})_2(^{13}\text{C}, ^{15}\text{N}\text{-MA})\text{Pb}_2\text{I}_7$ ($n = 2$). Moreover, only one strong excitonic absorption peak

can be observed in the absorption spectra of both 2D (BA)₂(MA)Pb₂I₇ (n = 2) and 2D (BA)₂(¹³C, ¹⁵N-MA)Pb₂I₇ (n = 2) as shown in Fig. S2. Therefore, the minor component (27.2 ppm peak) might more likely be structural impurity (eg. one PbI₆ octahedral unit missing or one spacer missing) according to recent work (*Nat. Nanotechnol* **15**, 969 (2020)). The amount of the minor MA component was estimated to be only 3% of the total amount of MA and its existence does not influence the conclusion of this work.

Revision made: The revision is described in the revision for the Q5 of the reviewer #2.

Q12. Line 227: as found in the crystal structure papers.

Answer: Thanks for the reviewer's suggestion. We followed the reviewer's suggestions. We reference structural papers (*J. Am. Chem. Soc* **1991**, 113, 2330-2332 and *Chem. Mater* **2016**, 28, 2852-2867) in Line 227.

Revision made: References have been modified in the revised manuscript.

Q13. Line 228-230: -- it does it by changing the crystal structure!

Answer: We totally agree with the reviewer's interpretation. Here, we answer this question together with Q14 below. Different types of organic spacers incorporated in 2D OIHPs may lead to different crystal structures, and subsequently cause the differences in molecular motion of the A-site cations.

Revision made: The following description is included in the discussion session to provide the argument.

“The structural deformation of the octahedral perovskite layers induced by the conformational change of the alkyl BA spacers in the 2D BA-based OIHPs occurs with cooling³². In contrast, there is no such structural change in both the inorganic framework of perovskites and the conformation of organic spacers in the 2D PEA-based OIHPs, evident from the ssNMR and PXRD analyses, respectively. Consequently, the structural deformation of the inorganic octahedral layers induced by changing the packing geometry of the BA organic spacers occurs on cooling, resulting in the change of the reorientational motion of the A-site cations. When the rigid spacer cation PEA was incorporated in 2D OIHP crystals, there is no such significant change neither in the

conformation of the spacer cation nor in the inorganic frameworks with cooling. Consequently, the reorientational dynamics of the A-site cation were found to stay relatively unchanged in the PEA-containing 2D OIHPs.”

Q14. Line 243: ‘... is spacer dependent’ -- again i.e. structure dependent.

Answer: We totally agree with the reviewer’s interpretation. Here, we answer this question together with Q13 above. Different types of organic spacers incorporated in 2D OIHPs may lead to different crystal structures, and subsequently cause the differences in molecular motion of the A-site cations.

Revision made: The revision is described in the response to Q13.

Q15. Line 254: The authors should mention the DSC result here, as it provides confirmation of the phase change and its temperature.

Answer: Thanks for the reviewer’s suggestion. We followed the reviewer’s suggestions to include the DSC results, as shown in Fig. S6. The endothermic peak occurred around 280K in the DSC measurement of 2D BA₂MAPb₂I₇ (n = 2) provides clear evidence of phase change.

Revision made: We included the DSC results as shown in Fig. S6 and pointed out that phase change of 2D OIHP BA₂MAPb₂I₇ (n = 2) occurred around 280K in the revised manuscript. In addition, the following description and the discussion of the DSC results have been included in the revised manuscript.

“Differential scanning calorimetry (DSC) and PXRD were further used to study the structural response of 2D (BA)₂(MA)Pb₂I₇ (n = 2) and 2D (PEA)₂(MA)Pb₂I₇ (n = 2) during the cooling process. The endothermic peak observed in the DSC measurement of 2D (BA)₂(MA)Pb₂I₇ (n = 2) indicated a clear phase change occurring at approximately 280 K (Fig. S6). This echoes the finding in a recent study³², showing the associated phase change was from Cmc₂m to P-1 space group. In contrast, no evidence of phase change was found in the DSC measurement of (PEA)₂(MA)Pb₂I₇ (n = 2). Moreover, based on the comparison of PXRD patterns recorded at 300K and 250K (Fig. 5(c) and 5(d)) a clear structural change for (BA)₂(MA)Pb₂I₇ (n = 2) was observed with cooling from 300K to 250 K, while no change was observed for the (PEA)₂(MA)Pb₂I₇ (n = 2).”

Q16. Line 259: conformational change – maybe, but might also be doubling of the content of asymmetric unit in the crystal.

Answer: Thanks for the reviewer’s comment. Our data interpretation is consistent with recent works (*Chem. Mater* **31**, 5592-5607 (2019) and *Chem. Mater* **33**, 3524-3533 (2021)). Regarding the responses of the materials to the temperature change, we agree that the BA-based 2D OIHP is going through a phase transition, as suggested by the reviewer, based on the NMR results together with the DSC and PXRD results.

Revision made: We have included the following text in the results section.

“This echoes the finding in a recent study³², showing the associated phase change was from Cmc₂m to P-1 space group.”

Q17. Line 270: How big is the shift?

Answer: The chemical shift of the major MA component in the 2D OIHP (BA)₂(¹³C, ¹⁵N-MA)Pb₂I₇ (n = 2) has shifted from 31.2 to 29.8 ppm.

Revision made: The chemical shift values have been provided in the revised supporting information. “The resonance peak of MA was found to have an upfield shift from 31.2 to 29.8 ppm with the temperature decreased from 308 to 243K.”

Q18. Line 286-287: It isn’t obvious how the author’s results show that the structural deformation of the PbI₆ layers – and again around line 327 .. There is kind of a circular argument here --- the single crystal structures will show variations in the similar, but not identical, PbI layers which affect the dynamics of the CH₃NH₃ ion enclosed within – so a change in MA dynamics indicates a change in its environment from one phase to another, but it does not tell you what that change in environment is.

Answer: The distortion of the octahedral PbI₆ unit due to the change in temperature have been reported based on single crystal X-ray diffraction data, as shown in recent work. (*Chem. Mater* **31**, 5592-5607 (2019))

Revision made: We have cited the relevant references to support our argument. Regarding the interpretation of the dynamics change of MA, we have answered it in our response to Q6.

Q19. Figure 1(b): there does not appear to be an ``overlap problem`` between the MA line and the organic spacer lines for the PEA compound! – but, granted, the isotopic labelling does greatly enhance the intensity.

Answer: Here, we would like to emphasize the clear overlap problem in the ^{13}C NMR spectrum 2D OIHP perovskite with BA spacer. Without tackling this problem, we will not be able to compare the materials made by BA spacer with the one made by PEA spacer. Yes, the signal intensity is greatly enhanced by using the isotopic labelling as mentioned by the reviewer.

Revision made: No additional revision made.

Q20. Figure 4: The caption mentions photoluminescence (PL) spectra but these are not in the figure – they have been moved to the supplemental section. I also note in Fig.4(d) that one of the ^{13}C aromatic peaks of the PEA reduces quite dramatically as T increases – probably a sign of reorientation about the phenyl axis.

Answer: Thanks to the reviewer for pointing out the inconsistency between the caption and the figure. Fig. 4 in the previous version is now Fig. 5 in the revised version. We have revised the caption of Fig. 5 and included the PL spectra in Fig. S7. We also thank the reviewer for pointing out that the dramatic PEA signal reduction as the temperature increase may be due to the phenyl ring reorientation about the phenyl axis.

Revision made: We have revised the caption of Fig. 5 and included the PL spectra in Fig. S7.

We have also included the following sentence in the revised manuscript.

“A signal reduction of a phenyl carbon resonance peak at 130 ppm with increasing temperature may be due to the flipping of the phenyl ring.”

Q21. “XRD” results:

The authors do not explain their “XRD” patterns (which ought to be labelled throughout as PXRD)– the series of equally spaced peaks have been indexed and these reflections relate to the long axis of the unit cell. For both compounds they could at least give the derived axis parameter and compare it with the single crystal information as this would

help confirm they have the right materials. They could also point out that these long axis parameters reflect the different spacings between the PbI₂ layers (which they would have found out earlier if they had looked at the single crystal papers).

Compare crystal parameters:

(BA)₂MAPb₂I₇ (ref 45): At 293K, Orthorhombic Cc2m, a=8.9470, b=39.347, c=8.8589, Z=4

(PEA)₂MAPb₂I₇ (Calabrese): At 294K, Triclinic P-1, a=8.794, b=8.792, c=22.766, α=94.02, β=97.02, γ=90.18, Z=2

From a rough measurement of the PXRD line spacings in figure S1 I calculate b=39.05 for the BA material and c=44.22 for the PEA material using their indexing numbers – note that the latter would correspond to a doubling of the unit cell given in the single crystal study (in line with the different Z values).

Answer: Thanks to the reviewer's suggestion to replace XRD with PXRD. Fig. 4 in the previous version is now Fig. 5 in the revised version. Our original assignments of the Miller planes of diffraction peaks, shown in Fig. 5, were based on the recent works (*Nano Lett* **17**, 4759-4767 (2017) and *Chem. Mater* **30**, 8538-8545 (2018)), considering 2D (PEA)₂MAPb₂I₇ to be of orthorhombic phase. It was also considered to be of triclinic phase in literature, as suggested by the reviewer (*J. Am. Chem. Soc* **113**, 2328-2330 (1991), *J. Phys. Chem. Lett* **11**, 6551-6559 (2020)). Among them, the single crystal data (*J. Phys. Chem. Lett* **11**, 6551-6559 (2020)) indicates that it is of triclinic phase. Thus, we have happily followed the reviewer's suggestion to define the Miller planes of diffraction peaks of 2D (PEA)₂MAPb₂I₇ structure using triclinic phase instead of orthorhombic phase. Accordingly, we estimated the lattice constant b for the BA-based material and the lattice constant c for the PEA-based material from Fig. S1. The lattice constant b for the BA-based material is calculated to be 39.24 Å, while the lattice constant c for PEA-based material is 22.07 Å. Our calculated results are consistent with the single crystal information, where the lattice constant b for the BA-based material and the lattice constant c for the PEA-based material are 39.34 and 22.76 Å, respectively, indicating that both the 2D (BA)₂MAPb₂I₇ and 2D (PEA)₂MAPb₂I₇ are successfully synthesized. We also like to thank the reviewer for pointing out the different spacings between 2D OIHPs with spacers BA and PEA. We can clearly observe that the long axis parameter of the PEA-based 2D OIHP is larger than the BA-

based 2D OIHP from Fig. S1, indicating that the choice of the incorporated spacer cation in 2D OIHP affects the structures of 2D OIHP.

Revision made: The Miller planes for 2D (PEA)₂MAPb₂I₇ in Fig. 5 have been reassigned based on the above explanations, changing from (020), (040), (060), (080), (0100), (0120) and (0140) to (001), (002), (003), (004), (005), (006) and (007). The new figure is shown below. We also replace the term “XRD” with “PXRD” in the revised manuscript.

Fig. 5 The ¹³C CPMAS spectra and PXRD measurement of 2D OIHP crystals ($n = 2$). The ¹³C CPMAS spectra of (a) 2D (BA)₂(¹³C, ¹⁵N-MA)Pb₂I₇ ($n = 2$) and (b) 2D (PEA)₂(¹³C, ¹⁵N-MA)Pb₂I₇ ($n = 2$) recorded at various temperatures, ranging from 308 to 243K. The PXRD patterns of (c) 2D (BA)₂MAPb₂I₇ ($n = 2$) and (d) 2D (PEA)₂MAPb₂I₇ ($n = 2$) measured at 300K and 250K.

Q22. Fig.S1: The PXRD patterns are not exactly the same as those shown in fig.4 of the main text – so are these at a different temperature, perhaps room temperature? Do they have an explanation for the different intensity patterns between the natural abundance and labelled samples?

Answer: Thanks for the reviewer's comment. Because the instrument used for the temperature-dependent PXRD measurements is different from the one used for the room temperature PXRD measurements (as shown in Fig.S1). As a result, the peak intensities of the two PXRD spectra are not exactly the same. The temperature-dependent PXRD measurement using a Bruker D8 Discover X-ray diffraction system with Cu-K α radiation in Bragg-Brentano geometry and equipped with a temperature control system (77-350K). Accordingly, we have revised the SM2 to include the details of the Materials Characterisations. Even though the intensities of the PXRD spectra of 2D OIHPs prepared with the natural abundance MA and $^{13}\text{C},^{15}\text{N}$ -MA are different, the positions of the PXRD patterns of these two samples are matched, indicating that the structure of 2D OIHPs ($n = 2$) did not alter due to replacing natural abundance MA with $^{13}\text{C},^{15}\text{N}$ -MA.

Revision made: The details of the Materials Characterisations have been revised in the SM2 to include the information of the instruments used for different PXRD measurements.

Q23. Fig. S3: For the BA compound going through the phase transition the shift increases by how much? What is the interpretation of the fairly substantial intensity decrease?

Answer: The chemical shift of the major MA component in the 2D OIHP $(\text{BA})_2(^{13}\text{C},^{15}\text{N}\text{-MA})\text{Pb}_2\text{I}_7$ ($n = 2$) has shifted from 31.2 to 29.8 ppm. We suppose that the reviewer meant that the intensity of the resonance peak of the incorporated MA in 2D OIHP $(\text{BA})_2(^{13}\text{C},^{15}\text{N}\text{-MA})\text{Pb}_2\text{I}_7$ ($n = 2$) decreased as the temperature increased. All the experiments in the various temperatures were performed with the same cross-polarization parameters. We believe the signal decrease was due to the cross-polarization efficiency change due to the change in the reorientational motion.

Revision made: The following sentence has been added in the caption of Fig. S5, which was Fig. S3 in the previously submitted version.

“The resonance peak of MA was found to have an upfield shift from 31.2 to 29.8 ppm with the temperature decreased from 308 to 243K.”

Q24. Fig. S4: The DSC (is this warming or cooling ?) very clearly shows a phase transition only for the BA compound, onset ~280K. This ought to be stated directly in the main text.

Answer: The DSC measurements in Fig. S6, which is the Fig. S4 in the version submitted previously, were recorded with the heating from 225 to 300 K. As the reviewer pointed out, obvious phase change is shown in the DSC heating curve of 2D OIHP BA₂MAPb₂I₇ (n = 2) in Fig. S6. Accordingly, we have revised our manuscript to include this data interpretation.

Revision made: We have included the experimental parameters of the DSC measurement in the SM2 as follows.

“The DSC heating curves were collected using a TA Q200 thermal analysis instrument at a scan rate of 5K min⁻¹ and heated from 225 K to 300 K in sealed aluminum pans under ambient conditions.”

In addition, the following text has been included in the revised manuscript to point out the occurrence of phase transition in the 2D OIHP BA₂MAPb₂I₇ (n = 2).

“The endothermic peak observed in the DSC measurement of 2D (BA)₂(MA)Pb₂I₇ (n = 2) indicated a clear phase change occurring at approximately 280 K (Fig. S6). This echoes the finding in a recent study³², showing the associated phase change was from Cmcm to P-1 space group. In contrast, no evidence of phase change was found in the DSC measurement of (PEA)₂(MA)Pb₂I₇ (n = 2).”

Q25. Fig. S5 caption uses (c) and (d) instead of (a) and (b) as in the figure itself. The spectra are for the 2D compounds without MA, and for the BA compound in (a) they ought to point out that there are likely two phase transitions evident from the changes in the C3 and C4 peaks.

Answer: We would like to thank the reviewer for pointing out the wrong labeling. The figure caption has been modified in the revised supporting information. Fig. S5 in the previous version is now Fig. S8 in the revised version. We agree with the reviewer’s interpretation about the changes in the C3 and C4 peaks of BA in the 2D OIHP BA₂PbI₄ (n = 1) crystal. Indeed, this is potentially phase transition evident, which echoes the

phase transition of the 2D OIHP BA₂PbI₄ (n = 1) reported mentioned in a previous work (*Acta Cryst.* **2007**, B63, 735-747).

Revision made: The figure caption has been modified in the revised supporting information and we have pointed out the phase transition in the 2D OIHP BA₂PbI₄ (n = 1) in the caption of Fig. S8 in the revised supporting information.

Q26. Fig. S6: The scale is barely legible. Assignments?

Answer: Thanks for the reviewer's comment. The figure is now Fig. S9 in the revised supporting information. We have added a more legible scale, assigning the proton signals, and adding a zoom insert that shows a two-bond ¹H-¹⁵N splitting of the methyl proton signals.

Revision made: We have replaced the figure with the new figure. In addition, the caption has been revised as follows.

“¹H NMR spectrum of ¹³C-methyl-¹⁵N-amine iodide in D₂O (850.3 MHz). The methyl group ¹H-¹³C *J*-coupling constant is 143 Hz.”

Fig. S9 ¹H NMR spectrum of ¹³C-methyl-¹⁵N-amine iodide in D₂O (850.3 MHz). The methyl group ¹H-¹³C *J*-coupling constant is 143 Hz.

Q27. Fig. S7 Presumably the splitting is from *J*-coupling to ¹⁵N, if so point this out.

Answer: Thanks for the reviewer's comment. This figure is now Fig. S10 in the revised supporting information.

Revision made: We have provided a clearer figure in the revised supporting information, as shown in Fig. S10. The figure caption has been revised accordingly as follows.

“Proton decoupled ^{13}C -NMR spectrum of ^{13}C -methyl- ^{15}N -ammonium iodide in D_2O (213.84 MHz). A ^{13}C - ^{15}N J-coupling of 6 Hz is observed.”

Fig. S10 Proton decoupled ^{13}C -NMR spectrum of ^{13}C -methyl- ^{15}N -ammonium iodide in D_2O (213.84 MHz). A ^{13}C - ^{15}N J-coupling of 6 Hz is observed.

Reviewer#3:

Comments:

The work by Lin et al. is a study of methylammonium dynamics in Ruddlesden-Popper lead halides based on phenylethylammonium (PEA) and butylammonium (BA). The authors use uniformly ^{15}N and ^{13}C labelled MA to investigate the dipolar coupling strength between the ^{15}N - ^{13}C spin pair in three different materials and as a function of temperature. The authors record REDOR data which they use to conclude that the

dynamics are qualitatively different in the PEA- and BA-based materials. The experimental data is of high quality. However, the analysis is relatively general and only focuses on qualitative differences between the MA dynamics in the different materials. That said, the authors have recorded quantitative REDOR data which should allow them to perform a more in-depth analysis to obtain quantitative information on the spatial restriction and correlation time of the MA motion. This analysis will render the work substantially more prominent and of interest to the broad readership of Nature Commun. working in the fields of halide perovskite optoelectronics and solid-state NMR. There are also a number of highly relevant references which have been overlooked by the authors and should be acknowledged. This reviewer therefore recommends that the work should be considered for publication after the authors have addressed the following queries:

“This result indicates that the MA in 2D $(\text{PEA})_2([\text{U-}^{13}\text{C}, \text{U-}^{15}\text{N}]\text{MA})\text{Pb}_2\text{I}_7$ ($n = 2$) underwent a more restricted reorientational motion than the MA in 2D $(\text{BA})_2([\text{U-}^{13}\text{C}, \text{U-}^{15}\text{N}]\text{MA})\text{Pb}_2\text{I}_7$ ($n = 2$), suggesting that the local environments for MA in these two materials were quite different.”

“The above results suggest that the reorientational motion of the A-site cations due to cooling in 2D OIHP crystals is spacer-dependent.

Q1. My main criticism regards the fact that the authors only draw qualitative conclusions regarding the MA dynamics (as illustrated by the above excerpts), despite having recorded high-quality data allowing them to perform quantitative analysis. The two key parameters characterizing dynamics are the order parameter and the correlation time, and the authors at present do not make an attempt to calculate and compare these parameters here for the three classes of materials under study.

Answer: We thank the reviewer’s comment and answer this question together with the Q8 raised by the reviewer #2. We took the suggestion to record the ^2H NMR spectra of 2D $(\text{BA})_2(\text{d}_3\text{-MA})\text{Pb}_2\text{I}_7$ ($n = 2$) and 2D $(\text{PEA})_2(\text{d}_3\text{-MA})\text{Pb}_2\text{I}_7$ ($n = 2$). The reorientational motion of MA provides motional averaging to the deuterium quadrupolar coupling of CD_3NH_3^+ , as well as to the dipolar coupling between the ^{13}C and ^{15}N spins of $^{13}\text{C}, ^{15}\text{N}$ -MA. Thus, we should be able to extract the information of the reorientational motion from both the ^2H NMR lineshape analysis and REDOR NMR. Our results indicated that these are the two complementary methods. We observed the multiple double-horned

quadrupolar splitting patterns in the ^2H NMR spectra, suggesting the existence of multiple motional modes of MA incorporated in 2D OIHPs. On the other hand, REDOR NMR, exhibiting the high resolution power of MAS, revealed the existence of the 3% of the minor MA component in 2D $(\text{BA})_2(^{13}\text{C}, ^{15}\text{N}\text{-MA})\text{Pb}_2\text{I}_7$ ($n = 2$).

Ideally, one should be able to construct a model to analyze both ^2H NMR line shapes and REDOR dephasing curve to extract the motional parameters quantitatively, provided the distribution of the C-N vector of MA relative to the PbI_6 octahedral unit is known. The different orientation preferences of MA may be associated with the different motional modes. However, unambiguous information is not available, preventing us from doing reliable analyses to extract the motional correlation times. The REDOR dephasing data at 2.4 ms can be correlated with the order parameters averaging over the different motional modes of the incorporated MA. The dynamics of the incorporated MA in BA- and PEA-based 2D OIHPs respond very differently to the temperature change. This main discovery is sufficiently supported with the provided evidence. We also like to thank the reviewer for pointing out some of the key references. They are all properly cited in the revised manuscript.

Revision made: We have included the suggested key references in the revised manuscript. These include the two references of the model-free approach^{53,54}, the REDOR derived order parameter, the previous studies of the spacer dynamics^{28,41,42} and relevant solid-state NMR works⁴³, listed below.

In addition to citing more references, we included the theoretical description of REDOR NMR in the presence of molecular motion to establish the relationship between the REDOR dephasing value and the order parameter, characterizing the reorientational motion. Moreover, we included the ^2H NMR experiments kindly suggested by the reviewer #2 for comparison. These changes are highlighted in our revised manuscript.

Reference

53 Lipari, G. & Szabo, A. Model-Free Approach to the Interpretation of Nuclear Magnetic Resonance Relaxation in Macromolecules. 1. Theory and Range of Validity. *J. Am. Chem. Soc* **104**, 4546-4559 (1982).

54 Haller, J. D. & Schanda, P. Amplitudes and Time Scales of Picosecond-to-Microsecond Motion in Proteins Studied by Solid-State NMR: A Critical Evaluation of

Experimental Approaches and Application to Crystalline Ubiquitin. *J. Biomol. NMR* **57**, 263-280 (2013).

28 Milic, J. V. *et al.* Supramolecular Engineering for Formamidinium-Based Layered 2D Perovskite Solar Cells: Structural Complexity and Dynamics Revealed by Solid-State NMR Spectroscopy. *Adv. Energy Mater* **9**, 1900284 (2019).

41 Ueda, T., Shimizu, k., Ohki, H. & Okuda, T. ^{13}C CP/MAS NMR Study of the Layered Compounds $[\text{C}_6\text{H}_5\text{CH}_2\text{CH}_2\text{NH}_3]_2[\text{CH}_3\text{NH}_3]_{n-1}\text{Pb}_n\text{I}_{3n+1}$ ($n = 1, 2$). *Z. Naturforsch. A* **51**, 910-914 (1996).

42 Dahlman, C. J. *et al.* Dynamic Motion of Organic Spacer Cations in Ruddlesden–Popper Lead Iodide Perovskites Probed by Solid-State NMR Spectroscopy. *Chem. Mater* **33**, 642-656 (2021).

43 Lee, J., Lee, W., Kang, K., Lee, T. & Lee, S. K. Layer-by-Layer Structural Identification of 2D Ruddlesden–Popper Hybrid Lead Iodide Perovskites by Solid-State NMR Spectroscopy. *Chem. Mater* **33**, 370-377 (2021).

Q2. In order for the dipolar coupling to be averaged out, the motion has to be isotropic (i.e. the order parameter has to vanish). When the authors refer to the motion as “more restricted” they imply that the order parameter is higher. Similarly, there is no attempt to calculate the correlation times. The discussion will benefit substantially from interpreting the results in the context of the model-free approach of Lipari and Szabo, see <https://pubs.acs.org/doi/10.1021/ja00381a009>. There are a number of experimental works using this framework for interpreting REDOR data (e.g. <https://link.springer.com/article/10.1007/s10858-013-9787-x>), which should greatly facilitate the task.

Answer: We would like to thank the reviewer for suggesting us to use the model-free approach to analyze the REDOR dephasing curve in the presence of molecular motion. Accordingly, we relate an order parameter linearly with the ratio of the motional averaged dipolar coupling constant to the dipolar coupling constant in the absence of motion, since both parameters share the same degree of randomness averaged by the same reorientational motion. Subsequently, we were able to derive the REDOR dephasing due to the motional averaged dipolar coupling between the ^{13}C and ^{15}N spins of $^{13}\text{C},^{15}\text{N}$ -MA and simulate the REDOR dephasing curves of the $^{13}\text{C},^{15}\text{N}$ -MA undergoing a different degrees of motions, characterized by different values of the order

parameters. In the absence of the reorientational motion, the order parameter is 1, the REDOR dephasing curve reaches the maximum value at 2 ms of the dipolar phasing time. In the presence of molecular reorientational motion, the order parameter will be between 1 and 0, and the REDOR dephasing value at 2 ms of the dipolar phasing time will be less than 1. The less the order parameter value, the less the REDOR dephasing value at 2 ms. Experimentally, we chose 2.4 ms of the dipolar dephasing time to compare so that XY-8 phase cycling can be applied. The statement is still true, namely that the less the order parameter value, the less the REDOR dephasing value at 2.4 ms.

Revision made: We have included the analysis of the REDOR dephasing due to the evolution of effective dipolar coupling, averaged by the molecular reorientational motion, characterized by the order parameter, in the revised manuscript. In addition to two pages of the descriptions in the results section, we also included Fig. S4, the simulated REDOR dephasing curves of the ^{13}C , ^{15}N -MA undergoing different degrees of reorientational motions, characterized by different values of the order parameters.

Q3. “Until now, only the dynamics of the organic spacers at the 2D OIHP crystals of $(4\text{NPEA})_2\text{PbI}_4$ and $(\text{PEA})_2\text{PbI}_4$ with $n = 1$, which consist of no A-site cations, have been analysed based on the temperature dependence of the ssNMR spectral line shapes.”

The authors should reference and acknowledge a study from 1996 which studied the dynamics of PEA in the $n=1$ material: <https://www.degruyter.com/document/doi/10.1515/zna-1996-0805/html>.

as well as more recent works, which reports solid-state NMR studies of spacer dynamics in various 2D OIHPs:

<https://onlinelibrary.wiley.com/doi/full/10.1002/aenm.201900284>. ($n = 1$ and $n = 3$ material based on FA)

<https://pubs.acs.org/doi/full/10.1021/acs.chemmater.0c03958>. ($n = 1$ materials for a wide range of different spacers)

Beyond dynamics, solid-state NMR studies of 2D OIHP have been rare so far with only a few reports so far. Acknowledging the above-mentioned works as well as the one below will provide a better representation of the current state of the art:

<https://pubs.acs.org/doi/abs/10.1021/acs.chemmater.0c04078>.

Answer: Thanks for the reviewer's suggestion. We have followed the reviewer's suggestion to add these four references in the introduction part of our manuscript.

Revision made: The last sentence of the introduction has been revised as follow to include the citation of these published works.

“The present study complements previous NMR studies focusing on the spacer molecules or the frameworks^{28,40-43}, and should provide new insights into the future design of 2D OIHP materials.”

Q4. In addition, please define “4NPEA”.

Answer: We have revised our manuscript to remove the description of 4NPEA.

Revision made: In the revised sentence, we only cited the reference of 4NPEA-related work but not mentioned it specifically in the sentence.

Q5. Regarding the assignment of the minor MA component in figure 1a: its chemical shift corresponds to unreacted MAI could the author report the two compounds on the same scale and comment? This assignment seems more plausible than invoking structural defects. It is also consistent with the higher CP efficiency (MA in MAI is rigid).

Answer: We have answered this question together with the Q5 and Q11 raised by the reviewer #2. We have plotted the ¹³C NMR spectra of 2D (BA)₂(¹³C, ¹⁵N-MA)Pb₂I₇ (n = 2) and the precursor ¹³C, ¹⁵N-MAI crystal powder on the same scale, as shown in Fig. S3. The possibility of the minor MA component being the unreacted ¹³C, ¹⁵N-MAI has been ruled out based on the difference observed in their ¹³C chemical shifts. In addition, the mixed phase of 2D OIHPs can easily be verified by powder X-ray diffraction (PXRD) characterization and absorption spectroscopy. (*Nat. Nanotechnol* **15**, 969 (2020)). Based on our results shown in Fig. S1, there are only one series of periodic repetitions of Miller planes without any additional PXRD patterns in both 2D (BA)₂MAPb₂I₇ (n = 2) and the one prepared with ¹³C, ¹⁵N-MA, 2D (BA)₂(¹³C, ¹⁵N-MA)Pb₂I₇ (n = 2). Moreover, only one strong excitonic absorption peak can be observed in the absorption spectra of both 2D (BA)₂(MA)Pb₂I₇ (n = 2) and 2D (BA)₂(¹³C, ¹⁵N-MA)Pb₂I₇ (n = 2) as shown in Fig. S2. Therefore, the minor component (27.2 ppm peak)

might more likely be structural impurity (eg. one PbI_6 octahedral unit missing or one spacer missing) according to recent work (*Nat. Nanotechnol* **15**, 969 (2020)). The amount of the minor MA component was estimated to be only 3% of the total amount of MA and its existence does not influence the conclusion of this work.

Revision made: The revision is described in the revision for the Q5 of the reviewer #2.

Q6. Please explain the notation: “U-“ (for uniform labelling)?

Answer: We answer this question together with the Q3 raised by the reviewer #2. [U- ^{13}C , U- ^{15}N] MA stands for uniformly ^{13}C and ^{15}N labeled MA. To ease the reading, the ^{13}C - and ^{15}N -enriched methylammonium is abbreviated as $^{13}\text{C},^{15}\text{N}$ -MA in the revised manuscript.

Revision made: The ^{13}C - and ^{15}N -enriched methylammonium is abbreviated as $^{13}\text{C},^{15}\text{N}$ -MA in the revised manuscript.

Q7. line 134: “sorely”

Answer: Thanks for the reviewer’s correction.

Revision made: The incorrect word “sorely” in line 134 has been replaced with the correct word “solely” in the revised manuscript.

Q8. line 135: “the growth of 2D OIHP crystals consisting of the conventional naturally abundant MA as the A-site cation, were also synthesised”

Answer: Thanks for the reviewer’s correction.

Revision made: In the revised manuscript, we removed this sentence and added another sentence as follow. “Here, we employed isotope labelling to distinguish A-site from spacer cations and avoid spectral overlap in 2D OIHPs with $n = 2$. Specifically, we employed $^{13}\text{C},^{15}\text{N}$ -MA and CD_3NH_3^+ as A-site cations, and investigated the dynamics of A-site molecular cations by REDOR NMR and ^2H NMR, respectively.”

REVIEWERS' COMMENTS

Reviewer #1 (Remarks to the Author):

The Authors have addressed my previous concerns and questions adequately in the revised manuscript. The addition of new experiments and broadening references help to put the present work in better context and affirm conclusions. They also improved the explanation of the REDOR results and their relevance to addressing the question of cation reorientation phenomena. I still had trouble making the connection between the NMR and fluorescence correspondence and the shifts mentioned are not clearly related to reorientational dynamics. However, this is an ancillary point and can be further explored in later work. Overall, I think the paper is publishable in its current form.

Reviewer #2 (Remarks to the Author):

Based on the reviewers' comments, the authors have made extensive revisions and additions to their manuscript which now is suitable for publications

Reviewer #3 (Remarks to the Author):

The authors have now addressed my queries in full. The results and their analysis are now of very high quality and fully warrant publication. The authors have also included additional data from complementary techniques which further strengthen the conclusions of their study.

The authors may want to consider referencing in their introduction a review article on using NMR to study dynamics in halide perovskites that appeared while their work was being revised:
<https://www.nature.com/articles/s41570-021-00309-x>

Response letter for “*Direct Investigation of the Reorientational Dynamics of A-site Cations in 2D Organic-Inorganic Hybrid Perovskite by Solid-State NMR*”

Previous Manuscript ID: NCOMMS-21-03696A

We have listed revisions to respond to the reviewers’ comments and suggestion listed point-to-point as follows.

Reviewer#1 (Remarks to the Author):

The authors have addressed my previous concerns and questions adequately in the revised manuscript. The addition of new experiments and broadening references help to put the present work in better context and affirm conclusions. They also improved the explanation of the REDOR results and their relevance to addressing the question of cation reorientation phenomena. I still had trouble making the connection between the NMR and fluorescence correspondence and the shifts mentioned are not clearly related to reorientational dynamics. However, this is an ancillary point and can be further explored in later work. Overall, I think the paper is publishable in its current form.

Answer: PXRD, NMR and PL spectroscopy are complementary techniques used to probe the structures, the chemical environments and the cation dynamics, and the optoelectronic properties of 2D OIHPs. We agree with the reviewer that there is still a room to explore the details of the connections among the observed phenomena. In the revised manuscript, we discuss the possible connection among these observed phenomena in the discussion section.

Revision made: We revised our data interpretation in the discussion section as follows.

“Consequently, the structural deformation of the inorganic octahedral layers induced by changing the packing geometry of the BA organic spacers occurs on cooling, resulting in the change of the reorientational motion of the A-site cations and the change of optoelectronic property indicated by the shift in the PL spectrum.”

Reviewer#2 (Remarks to the Author):

Based on the reviewers' comments, the authors have made extensive revisions and additions to their manuscript which now is suitable for publications.

Answer: We are grateful that the reviewer provided valuable comments for the first-round reviewing process, helping us to revise this manuscript.

Reviewer#3 (Remarks to the Author):

The authors have now addressed my queries in full. The results and their analysis are now of very high quality and fully warrant publication. The authors have also included additional data from complementary techniques which further strengthen the conclusions of their study.

The authors may want to consider referencing in their introduction a review article on using NMR to study dynamics in halide perovskites that appeared while their work was being revised: <https://www.nature.com/articles/s41570-021-00309-x>

Answer: We would like to thank the reviewer for suggesting us to cite this recent review article, which discusses extensively about the application of solid-state NMR in probing microstructure, dynamics and doping of metal halide perovskites.

Revision made: We have revised the penultimate sentence of the first paragraph of the introduction to include this article as a reference #20.

“In particular, ssNMR has emerged as a useful tool for studying OIHP²⁰ and the cation reorientational dynamics in OIHP.”